# Mobile atmospheric measurements and local-scale inverse estimation of the location and rates of brief CH$_4$ and CO$_2$ releases from point sources

Pramod Kumar[1], Grégoire Broquet[1], Camille Yver-Kwok[1], Olivier Laurent[1], Susan Gichuki[1], Christopher Caldow[1], Ford Cropley[1], Thomas Lauvaux[1], Michel Ramonet[1], Guillaume Berthe[2], Frédéric Martin[2], Olivier Duclaux[3], Catherine Juery[3], Caroline Bouchet[4], Philippe Ciais[1]

[1]Laboratoire des Sciences du Climat et de l'Environnement (LSCE/IPSL), CEA-CNRS-UVSQ, Université Paris-Saclay, 91191 Gif-sur-Yvette, France
[2]IFP Energies nouvelles-Géoscience, 92852 Rueil-Malmaison Cedex, France
[3]TOTAL Laboratoire Qualité de l'Air (LQA), 69360 Solaize Cedex, France
[4]SUEZ-Smart & Environmental Solutions, Tour CB21/16 place de l'Iris, 92040, La Défense, France

*Correspondence to*: Pramod Kumar (pramod.kumar@lsce.ipsl.fr)

## Abstract

We present a local-scale atmospheric inversion framework to estimate the location and rate of methane (CH$_4$) and carbon dioxide (CO$_2$) releases from point sources. It relies on mobile near-ground atmospheric CH$_4$ and CO$_2$ mole fraction measurements across the corresponding atmospheric plumes downwind of these sources, on high-frequency meteorological measurements, and on a Gaussian plume dispersion model. The framework exploits the scatter of the positions of the individual plume cross-sections, the integrals of the gas mole fractions above the background within these plume cross-sections and the variations of these integrals from one cross-section to the other to infer the position and rate of the releases. It has been developed and applied to provide estimates of brief controlled CH$_4$ and CO$_2$ point source releases during a one-week campaign in October 2018 at the TOTAL's experimental platform TADI in Lacq, France. These releases lasted typically 4 to 8 minutes and covered a wide range of rates (0.3 to 200 gCH$_4$/s and 0.2 to 150 gCO$_2$/s) to test the capability of atmospheric monitoring systems to react fast to emergency situations in industrial facilities. It also allowed testing their capability to provide precise emission estimates for the application of climate change mitigation strategies. However, the low and highly varying wind conditions during the releases added difficulties to the challenge of characterizing the atmospheric transport over the very short duration of the releases. We present our series of CH$_4$ and CO$_2$ mole fraction measurements using instruments onboard a car that drove along roads ~50 to 150 m downwind of the 40 m × 60 m area for controlled releases along with the estimates of the release locations and rates. The comparisons of these results to the actual position and rate of the controlled releases indicate ~10%-40% average errors (depending on the inversion configuration or on the series of tests) in the estimates of the release rates and ~30-40m errors in the estimates of the release locations. These results are shown to be promising especially since better results could be expected for longer releases and under meteorological conditions more favorable to local scale dispersion modeling.

However, the analysis also highlights the need for methodological improvements to increase the skill for estimating the source locations.

## 1. Introduction

Accurate detection and quantification of greenhouse gas (GHG) emissions from anthropogenic activities is essential to construct effective mitigation policies. A large fraction of pollutant and greenhouse gases
comes from industrial sites. Between 30% and 42% of the anthropogenic emissions of methane ($CH_4$) between 2008 and 2017 are from the fossil fuel production and use sector (coal, natural gas and oil) according to Saunois et al. (2019). A recent study by Hmiel et al. (2020) suggests that anthropogenic fossil $CH_4$ emissions have been underestimated by about 38 to 58 Tg/year, which could implicitly rise the contribution of this sector by 25%-40%. $CH_4$ emissions inventories for specific sectors combine uncertain
activity data and highly uncertain emission factors (Alvarez et al., 2018). Furthermore, typical emission factors used as the default values in inventories can hardly be representative of the specific configurations and processes of individual sites, and, in practice, they are usually different from those measured at specific sites (e.g. Vaughn et al., 2017; Ravikumar et al., 2017; Omara et al., 2018) Monitoring of $CH_4$ emissions from individual sites and even at the scale of local facilities within the same site is thus
recommended to assess the effectiveness of local measures applied to minimize emissions (Konschnik et al., 2018).

$CH_4$ emissions from industrial activities are often strongly localized and can occur at many places with all kinds of frequencies or temporal scales (continuous to infrequent, constant, highly variable) (Zavala-Araiza et al., 2018). $CH_4$ can be emitted at various stages of activities related to oil and gas production,
transport and use, such as from venting during oil extraction, pressure controllers, unintended fugitive emissions across the entire process chain, pressure regulators along distribution through pipelines, and storage (Höglund-Isaksson, 2017). Some of these emissions could be localized through periodical LDAR (Leak Detection and Repair) campaigns. Such $CH_4$ emissions are often accompanied by $CO_2$ emissions, for example when considering diesel engines powering large compressors or flaring activities to reduce
natural gas (NG) venting (Caulton et al. 2014). Therefore, the monitoring of $CO_2$ emissions whose budget can be significant and which can help detect and characterize the processes underlying the $CH_4$ emissions is important too.

For Oil and Gas (O&G) related activities, fugitive emissions, for example from leaky valves or air bleeds from compressors, should be distinguished from intermittent emissions that occur during nominal and
65 maintenance operations like purging and draining of pipes. Several recent studies have shown that a few leaks, often referred to as super-emitters, can be responsible for a large fraction of the O&G emissions of a site, creating a long-tail distribution of emission sources (Omara et al., 2016; Zavala-Araiza et al., 2015, 2017; Frankenberg et al., 2016; Alvarez et al., 2018). Therefore, reducing infrequent but large releases of $CH_4$ is an effective strategy for reducing the overall emissions of the entire O&G sector (Duren

et al, 2019). In addition to their effect on climate, large sporadic $CH_4$ emissions can also be an issue for safety, a further argument for developing and deploying fast detection and quantification systems.

Atmospheric $CH_4$ and $CO_2$ mole fraction measurements in the vicinity of industrial sites, or of facilities within a site, have been used for detecting, localizing and quantifying local emissions. These data are combined with tracers or atmospheric transport models for the localization of sources, and dual tracer methods, mass balance approaches or atmospheric transport inverse modelling techniques to quantify release rates (Foster-Wittig et al., 2015; Albertson et al., 2016; Ars et al., 2017; Yacovitch et al., 2017; Feitz et al., 2018; etc.). Current measurement methods include both in situ and remote sensing measurements from fixed stations or mobile platforms (with instruments onboard aircraft, automobile, or drones) (Peischl et al., 2013; Pétron et al., 2014; Brantley et al., 2014; Goetz et al., 2015; Foster-Wittig et al., 2015; Albertson et al., 2016; Alvarez et al., 2018; Feitz et al., 2018; Cartwright et al., 2019, etc.). Controlled release experiments have been regularly conducted to support the development, test and improvement of atmospheric measurement and modeling techniques for the detection, localization and quantification of emissions (Loh et al., 2009; Lewicki and Hilley, 2009; Ro et al., 2011; Humphries et al., 2012; Kuske et al., 2013; van Leeuwin et al., 2013; Luhar et al., 2014; Foster-Wittig et al., 2015; Jenkins et al., 2016; Hirst et al., 2017; Ars et al., 2017; etc.).

TOTAL developed the so-called TOTAL Anomaly Detection Initiatives (TADI) platform at Lacq in southwestern France as a test bed for different GHG measurement technologies and emission detection and quantification methods that could be implemented to support either the fast detection of large leaks or the estimate of the long-term budget of the GHG emissions from facilities. On this TADI platform, a wide-range of industrial equipment (pipes, valves, tanks, columns, wellhead, flare, etc.) are used to reproduce around 30 different leaks scenarios including the most likely to occur on operational sites (cold venting, leaks from a flange, leaks from a connection, leakage of valves, leakage under insulation, corrosion on a line, etc.). In October 2018, a one-week campaign was held at the TADI platform to evaluate different approaches to determine the precise location and magnitude of brief $CH_4$ and $CO_2$ controlled releases from point sources. Different groups with various atmospheric measurement and modelling techniques participated in the campaign. With typically 4-8-minute releases, the experiment was mainly designed for testing safety surveillance systems addressing emergency situations rather than for testing the ability to quantify routine emissions accurately over long periods of time. However, a wide range of rates were used for the controlled releases, including large releases that can raise safety issues but also small releases, which mainly raise concerns for climate change. Such a wide range of sporadic releases was a challenge for the systems deployed by the participants since they required highly precise gas analyzers that operate at both low and high atmospheric gas mole fractions, and the analysis of atmospheric processes over short durations.

We participated in this campaign within the framework of the TRAcking Carbon Emissions (TRACE) program (https://trace.lsce.ipsl.fr/), using a mobile measurement strategy similar to that of Yver Kwok et al. (2015) and Ars et al. (2017), with the Cavity Ring Down Spectrometers (CRDS) instruments onboard of a vehicle driven back and forth across $CH_4$ and $CO_2$ plumes to get as many cross-section measurements as possible for each release. The measurements were made along roads downwind of the TADI platform with the air intake located ~2 m above the ground. Such mobile measurements are generally conducted occasionally, and they are hardly adapted to continuous long-term screening for the fast detection of dangerous leaks. However, such measurements could be conducted regularly to get a representative diagnostic of emissions from a site and of their evolution with time. Furthermore, the development of automated mobile platforms with light instruments could allow for the use of such a measurement strategy for long-term systematic monitoring of the emissions from a site.

Such mole fraction measurements near the ground and across the plume from the source are often coupled to the release of a tracer gas at a known rate close to a targeted source in order to quantify the corresponding emission by exploiting the mole fraction ratios between the targeted gas and the tracer (Yver Kwok et al., 2015). However, one can hardly conduct such tracer releases over long time periods or within areas exposed to safety issues. Furthermore, using this method it is difficult to localize the targeted source since the method itself relies on a good knowledge of the source position. The use of dispersion models to analyze mobile near ground data for the estimation of source locations and rates can be challenging (Foster-Wittig et al., 2015; Ars et al. 2017). Furthermore, most of the atmospheric inversion approaches to localize and quantify point sources have been developed and tested for releases lasting ~30 min or more (Feitz et al., 2018) whereas the TADI releases during this campaign did not exceed 18 minutes. Because of the short duration of those releases, only a small number of plume cross-sections could be obtained for each release, limiting the robustness of the inversions. Finally, the meteorological conditions during the campaign were quite challenging, with low wind speed and highly varying wind directions. We had to develop a specific and pragmatic inversion approach to overcome these challenges, exploiting the spread of the positions of the few individual plume cross-sections, the integrals of the mole fraction above the background (i.e. the level of gas mole fraction behind that of the plume from the targeted source that is due to remote sources and sinks) within these plume cross-sections, and the variations of these integrals from one cross-section to the other in order to infer the position and rate of the brief releases. This inversion approach is based on a Gaussian plume model whose parameters were fixed using the meteorological measurements conducted on the TADI platform. Its successful retrieval of relatively good release rates confirm that it could feed more advanced strategies for the local scale monitoring of GHG emissions.

This study documents our measurements, analysis, inversions and the comparison of the results to actual release location and rates during the TADI-2018 campaign. In section 2, we detail the experimental setup and atmospheric measurements. The theoretical and computational frameworks of the inversion approach

are described in section 3. Section 4 details the data analysis for the configuration of the transport model and of the inversion. The results and perspectives of the study are discussed respectively in sections 5 and 6, followed by the conclusions in section 7.

## 2. The TADI-2018 campaign

### 2.1 The site, controlled releases and atmospheric conditions

The TADI-2018 campaign was conducted during October 15-19, 2018 at TOTAL's TADI platform in Lacq, northwest of Pau. The platform is a rectangular area of approximately 20000 $m^2$ with decommissioned oil and gas equipment installed to mimic typical equipment of a "real-world" oil and gas facility. Within the platform, there are different points from which $CH_4$ and / or $CO_2$ can be released at controlled rates from low (e.g. few tens of $gCH_4/s$ or $gCO_2/s$) to relatively high (e.g. several hundreds of
$gCH_4/s$ or $gCO_2/s$). There are chemical and industrial plants to the East of the platform, and the surrounding area has agricultural land and rural settlements. The terrain of the TADI platform is almost flat. However, during controlled release experiments, there were small obstacles to the atmospheric dispersion: tents covering the instruments, the decommissioned oil and gas equipment, and other small infrastructure for storage create which increased the roughness and inhomogeneity of the TADI platform.
Figure 1 shows a schematic of our experimental setup during the TADI-2018 campaign.

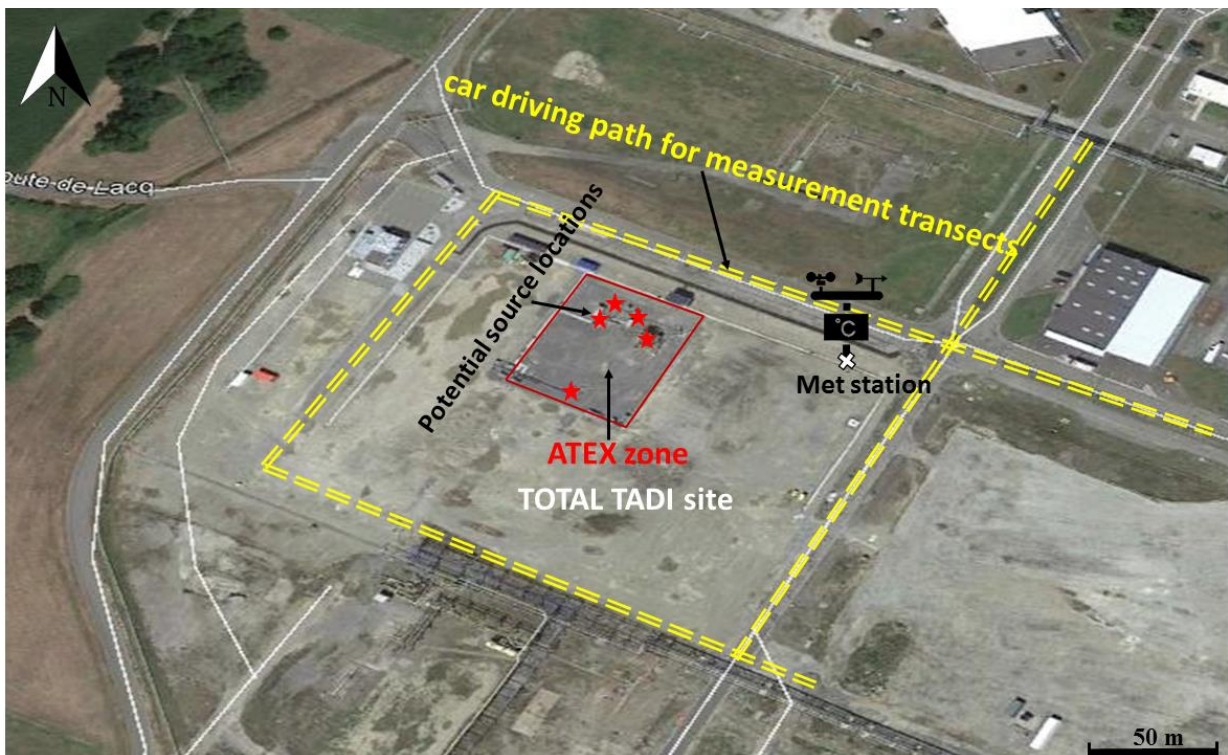

Figure 1: A schematic of the experimental setup on the top of the satellite image of the TADI platform (source: Google Earth © Google Earth). The red stars show some of the possible approximate location of the emission sources in the ATEX zone (rectangle with red colored line). The full set of exact locations used for the releases is detailed in Figure S1 of the supplementary information (SI) material. A hybrid SUV drove in electric mode on the road next to the site, along the yellow colored double dotted lines. The meteorological station installed and operated by TOTAL was located at the basis of its black symbol.

During the campaign, a total of 50 $CH_4$ and $CO_2$ releases were carried out. All these controlled releases were made from different point source locations within a 40 m × 60 m rectangular area classified as the "ATEX zone" (Figures 1 and S1, in the supplementary material), which for security reasons was cordoned off and out of reach for all participants. These point sources correspond to various types of equipment and release scenarios: drilled plugs, pipes, rack corrosion, flanges, valves, control boxes, horizontal or vertical tubing, horizontal or vertical piping, manhole, under insulation, tanks, scrubbers, product skids (red stars in Figures 1 and S1) with different release heights between 0.1 m and 6.5 m above the ground. Mass flow controllers were used to control the releases of $CH_4$ and $CO_2$. Several series of releases were performed with pauses of approximately 5 minutes between two releases and with a range of emission rates varying from 0.3 $gCH_4/s$ to 200 $gCH_4/s$ for $CH_4$ and from 0.2 $gCO_2/s$ to 150 $gCO_2/s$ for $CO_2$. This setup allowed the reproduction of a variety of gas release scenarios expected in an industrial environment.

## 2.2 Atmospheric measurements

Atmospheric $CH_4$ and $CO_2$ mole fractions were measured using two Picarro CRDS: Picarro G2203 and G2401 analyzers for $CH_4$ and $CO_2$, respectively. The analyzers were calibrated at the beginning and end of the experiment using high and low range calibration standards traceable to the WMO scales (WMOX2007 for $CO_2$, and WMOX2004A for $CH_4$; WMO GAW report No. 242; Table 1). Each standard was measured for at least 20 minutes on each analyzer. The agreement errors between the analyzer raw data and the calibration standard were smaller than 0.7% in $CO_2$ and 0.2% in $CH_4$. Yver Kwok et al. (2015) had shown that within the mole fraction range of the WMO scales the analyzer precision of an ensemble of CRDS analyzers including the G2401, defined as the raw data standard deviation over one minute, was <0.05 ppm and <0.5 ppb for $CO_2$ and $CH_4$, respectively. The G2203 analyzer is based on the same spectroscopy as the CRDS analyzers investigated in this study. It was tested in a similar way during S. Ars PhD study and displayed similar performance (Ars, 2017). CRDS instruments are known to be stable within <0.15 ppm per year for $CO_2$ and <2.2 ppb per year for $CH_4$ (Yver Kwok et al., 2015).

Table 1: Assigned mole fraction of calibration standards used during the campaign; SD refers to the calibration reproducibility, which is defined as the standard deviation (1σ) of the means of at least 3 independent measurements.

|         | $CO_2$ (ppm) | $CO_2$ SD (ppm) | $CH_4$ (ppb) | $CH_4$ SD (ppb) |
|---------|--------------|-----------------|--------------|-----------------|
| High    | 522.25       | ±0.01           | 6135.03      | ±0.23           |
| Low     | 411.94       | ±0.01           | 1980.65      | ±0.11           |

During the campaign the range of measured mole fractions corresponding to the releases selected for the inversions (see section 4.2) was 1.9 – 84 ppm for $CH_4$ and 400 – 800 ppm for $CO_2$, with less than 4% of the $CH_4$ measurements and less than 2% of the $CO_2$ measurements being higher than the CRDS calibration range shown in Table 1. The manufacturer specifications recommend operating ranges of 0-20 ppm for
$CH_4$ and 0-1000 ppm for $CO_2$ with the G2203 and G2401 analyzers, respectively. In practice the analyzers were still operational over a higher range although lower performance may be expected in this case. To investigate the performance of both analyzers at high mole fractions, a test of linearity was conducted at the Laboratoire des Sciences du Climat et de l'Environnement (LSCE) over a range of mole fractions of 2 - 50 ppm for $CH_4$ and 400 - 5000 ppm for $CO_2$, which spans ~99% of the $CH_4$ measurements recorded
during the releases selected for the inversions. The results indicate that over this range, the precision was < 20 ppb for $CH_4$ and < 0.6 ppm for $CO_2$ with the G2203 and G2401 analyzers, respectively, and that both analyzers still responded linearly ($R^2 > 0.99$) at high mole fractions, with a residual errors between the gas analyzer responses and the assigned values lower than 2%.

The gas analyzers were installed in a Mitsubishi hybrid SUV vehicle. Measurements were made
continuously at approximately 0.3-0.4 Hz while the vehicle was driven up and down the two main roads next to the TADI platform at a speed of about 10 km/h (which resulted in getting ~1 measurement every 7m) (Figure 1). The distance between the release points and the car was between ~25 m and about ~250 m. Due to the brevity of the releases, less than six cross-sections of the plume were identified in the mobile transects for each controlled release. The sampling inlet was located at the back of the vehicle, at
approximately 2 m above the ground. The top of the sampling mast was equipped with a GPS providing a time reference along with measurement positions. At the beginning of the campaign, the overall time delay of the different analyzers, including the time delay induced by the sampling line and the analyzer time shift relative to GPS time, was empirically assessed by contaminating (breathing out) shortly at the air inlet at a given GPS time and comparing this time to the analyzer timestamp of the $CO_2$ response (at
peak summit). The measurements were thus synchronized with an overall time delay of 16 s. Figure 2 shows an example of the transects on the TADI adjacent roadways, with the timeseries of observed instantaneous $CH_4$ mole fractions during a $CH_4$ release.

In the absence of a controlled tracer release, reliable measurements of the meteorological and turbulence parameters are essential to model the plumes from the releases with an atmospheric dispersion model. A
meteorological station was installed and operated by TOTAL in the north-east of the ATEX zone (Figure

1). This station included a Metek Sonic 3D sonic anemometer at 10 m height above the ground. The high frequency measurements of this anemometer were not recorded but combined at 1-minute resolution into mean horizontal wind speed ($U$) and direction ($\theta$), temperature (T), Obukhov length ($L$), surface friction velocity ($u_*$), and standard deviation of wind velocity fluctuations ($\sigma_u$, $\sigma_v$, and $\sigma_w$). We averaged these 1-minute meteorological data over the entire release periods and used these as inputs for the modelling and inversion configurations. Therefore, the notations $U$, $\theta$, T, $L$, $u_*$ and ($\sigma_u$, $\sigma_v$, $\sigma_w$) hereafter represent such averages over the release periods rather than the 1-minute data. All the releases were conducted during daytime under near-neutral or convective stability conditions ($L < 0$). The prevailing atmospheric conditions during the whole campaign corresponded to low and highly variable south-west to south-east winds.

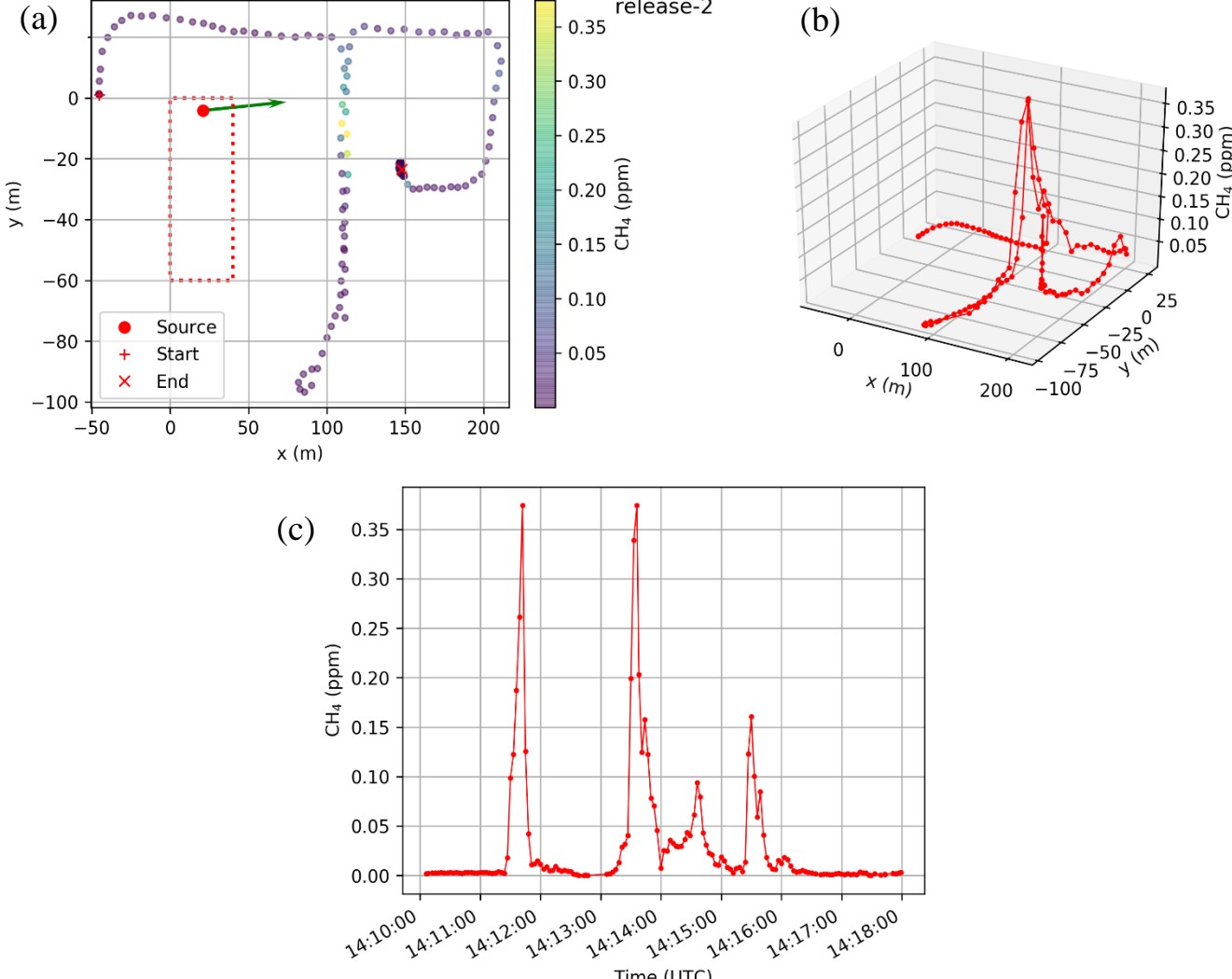

Figure 2: Mobile $CH_4$ mole fraction measurements during $CH_4$ release no. 2 (Table 2): (a) horizontal representation (b) 3D representation with values as a function of the horizontal location, and (c) time series. The green arrow from the source location in (a) shows the averaged wind direction during that release.

## 3. Atmospheric inversion of the release locations and rates

### 3.1 Gaussian plume dispersion model

The atmospheric inversion approach used here relies on a Gaussian plume model to simulate the dispersion of $CH_4$ or $CO_2$ from the potential locations of the sources. Gaussian plume models (Hanna et al., 1982) provide an approximation of the average tracer dispersion at a local scale (for source-receptor distances of less than a few kilometers) driven by constant meteorological conditions in time and space over a flat area. In such conditions, the concentration ($C$) of a pollutant has a spatial distribution described by a combination of normal distributions in both vertical and horizontal planes (Hanna et al., 1982). We use the following Gaussian model formulation assuming a reflective ground surface:

$$C(X,Y,Z) = \frac{Q_s}{2\pi\sigma_Y\sigma_Z U_e} exp\left(\frac{-Y^2}{2\sigma_Y^2}\right)\left[exp\left(\frac{-(Z-z_e)^2}{2\sigma_Z^2}\right) + exp\left(\frac{-(Z+z_e)^2}{2\sigma_Z^2}\right)\right] \quad (1)$$

where the $X$ and $Y$ axis are defined by the effective wind direction, $Q_s$ is the emission rate of the point source underlying the plume, $z_e$ is the effective release height above the ground surface, $U_e$ is the effective mean wind speed at the height of the release, ($X$, $Y$, $Z$) are the coordinates in the Gaussian model concentration space where the effective location (accounting for the effective injection height, Briggs, 1975) of the release is (0,0, $z_e$) (this system of coordinates is distinct from the coordinate system used in the following sections to localize the sources in the ATEX zone), and $\sigma_Y$ and $\sigma_Z$ are the dispersion coefficients in lateral ($Y$) and vertical ($Z$) directions, respectively. The dispersion coefficients $\sigma_Y$ and $\sigma_Z$ are derived from the standard deviations of the corresponding velocity fluctuations in the lateral ($\sigma_v$) and the vertical ($\sigma_w$) directions as follows (Gryning et al., 1987):

$$\sigma_Y = \sigma_v t \left(1 + \sqrt{\frac{t}{2T_Y}}\right)^{-1} \quad (2a)$$

$$\sigma_Z = \sigma_w t \left(1 + \sqrt{\frac{t}{2}}\right)^{-1} \quad (2b)$$

where $t$ ($= X/U_e$) is the travel time from origin to $X$, $T_Y$ and $T_Z$ are the Lagrangian time scales in lateral ($Y$) and vertical ($Z$) direction, respectively. We take $T_Y = 200$ s (Draxler, 1976) for near surface release and $T_Z = 300$ s for unstable conditions ($L < 0$) (Gryning et al., 1987).

The TADI platform is relatively flat and we assume that the small obstacles interfering with the plumes between the ATEX zone and our measurement locations are negligible, which is the main reason for using a Gaussian model here. Furthermore, our inversion method relies on a very high number of plume simulations to localize the sources, which was not affordable with complex models. Advantages of more complex models like the ability to account for variations in space and time of the wind were challenged by the very short duration of the releases, which prevented us from considering such variations. We also had to rely on a single meteorological station which limited the skill to account for spatial variations in the wind. The prevailing wind conditions during the whole campaign with low wind speeds and highly variable wind directions challenged the spatial representativeness of the meteorological measurements and the use of local-scale dispersion models to simulate the peaks in the mobile measurement transects. Such a limitation applies to Gaussian models as well as to more complex models although our inversion approach attempts to take advantage of strong variations in the wind direction to localize the sources.

The small number of plume cross-sections (also called "peaks" hereafter) observed in this study prevented us from assessing the average mole fractions along the roads where mobile measurement transects were conducted for each release. The average in time of the gas mole fractions measured along all roads is far from converging towards a distribution corresponding to an average plume and just reflects the scattering of these peaks. However, even though a Gaussian model characterizes average plumes under constant wind and it can thus substantially deviate from observed instantaneous mole fractions, we compared plume cross-sections simulated with such a model to the observed instantaneous plume cross-sections. We consider the integral of the mole fractions above the background within cross-sections as the index of the plume amplitude whose observed value is fitted by the model in the inversion approach, which limits the impact of the lack of simulation of the turbulent patterns (Monster et al., 2014; Alberston et al., 2016; Ars et al, 2017). With such a framework, the Gaussian model was assumed to be suitable to assimilate the information from our instantaneous plume cross-sections, which was confirmed to a large extent by the precision of the release rate estimates from the inversion based on this model (see sections 5 and 6). Furthermore, the model error associated with such a use of the Gaussian model to simulate instantaneous plume cross-sections is implicitly accounted for in the inversion configuration (see section 3.2). Using advanced and more complex models simulating explicitly the turbulence to help better match observed instantaneous plume cross-sections could be considered as a next step but this raises challenges since it is difficult to capture the right timing and location of turbulent stochastic structures. Despite many attempts at developing systems based on complex models, in practice, the systems used for the local scale monitoring of $CH_4$ emissions generally rely on mass balance approaches or Gaussian models (Fox et al, 2019; Mønster et al, 2019)

## 3.2 Inversion method

The inversion system primarily relies on the plume amplitudes (defined as the integral of the gas mole fractions above the background in peaks as in Ars et al. (2017); see section 3.1) along the mobile measurement transects to infer the release rates. These amplitudes are the main component of the data assimilated by the inversion system. They highly depend on both the release rate and the distance from the source, whose location is unknown, to the measured peaks. Indeed, the plume amplitude at the measurement height is smaller at larger distance because (i) of the plume larger vertical mixing and (ii) of the loss of larger tails of the plume in the integration when the plume is wider and smoother and its concentrations get closer to the background. These amplitudes also depend on the angle between the plume cross-sections and the effective wind directions from the source to these cross-sections, which provides another sensitivity to the source location. The inversion scheme also follows the fact that, due to unsteady wind conditions and turbulence, the effective wind directions from the release point to the peaks in the mobile measurement transects along the roads can differ from $\theta$, the mean wind direction averaged over the brief release periods. However, the variability of the wind measurements at high frequency should give a good indication of the fluctuations of such effective wind directions. This provides information about the source location so the position of the peaks along the mobile measurement transects are the other component of the data assimilated by the inversion system. Crossing the information about the varying amplitude of the different peaks and about their location adds a critical piece of information about the source location, since the variations of the effective wind from a source to the roads strongly impact the distance between the source and the peaks and the angle between the effective wind and the plume cross-section, and thus, the peak amplitudes. The analysis of the variations of the different peak amplitudes is necessary to disentangle the estimates of the rate and location of a release, since changes in the average peak amplitude due to changes in the release location can be compensated by change in the release rate. Therefore, our method relies on the information from multiple plume cross-sections to infer unambiguously both the rate and location of the releases.

In practice, in order to compare modeled peaks to measured ones, the inversion drives the Gaussian model with an effective wind direction $\theta m$, but with an effective wind speed and plume widths that are constrained with the meteorological measurements. $\theta m$ is defined by the direction between the potential source locations and the peak locations. More specifically, $\theta m$ is taken as the direction from the potential source location to the "center" of the measured peak. This center is estimated as the mid-point between the edges of the measured plume cross-sections, these edges being defined manually. The confidence in the $\theta m$ corresponding to a given source location is weighted by its relative departure from $\theta$ compared to $\sigma_\theta$, the standard deviation of the measured wind direction over the release period. Since the high frequency measurements of the wind were not recorded, for each release $\sigma_\theta$ is approximately calculated as $\sigma_\theta \simeq \sigma_v/U$ (Joffre and Laurila, 1987).

The Gaussian model driven by these parameters yields a simulation of the 3D field of mole fractions above the background due to the source. This 3D field of mole fractions is discretized at the measurement locations. The observed $Ao$ and modeled $Am$ plume amplitudes are computed as integrals along these locations of the mole fractions above the background between the edges of the observed peak. These edges are defined manually, and the derivation of the background in the observations is detailed in section 4.1.

We provide $z_s$ the actual source height of each release to the inversion system, which assumes that the effective injection height $z_e$ corresponds to this height ($z_e=z_s$). The inversion derives estimates of the horizontal source location, knowing it is within ATEX zone, but ignoring any information about the set of actual source locations listed in Figure S1. It discretizes the ATEX zone into small cells of 1 m$^2$ to define all potential horizontal locations ($x$, $y$) of the source. For each controlled release, the inversion algorithm loops over all these locations and on an extensive ensemble of values for the release rates $Q$ with intervals of 0.05 gX/s (or of 0.1 gX/s if measurements at first sight indicate that the emission rate is likely well above 10 gX/s; where X=CH$_4$ or CO$_2$) to find the optimal estimates of the release location and rate. For each potential location and rate, it drives one Gaussian plume simulation per plume cross-section following the principle detailed above, and computes the corresponding amplitudes of the modeled plume cross-sections. Then it computes the corresponding cost function $J$ defined by:

$$J = J_p + J_w \qquad (3)$$

where the first term:

$$J_p = \sum_{i=1}^{N_p} \left[\frac{Ao_i - Am_i}{Ao_i}\right]^2 \qquad (4)$$

is the quadratic sum of relative errors between the modeled ($Am_i$) and observed ($Ao_i$) amplitudes of the $N_p$ plume cross-sections and the second term:

$$J_w = \sum_{i=1}^{N_p} \left[\frac{\theta - \theta m_i}{\sigma_\theta}\right]^2 \qquad (5)$$

is the quadratic sum the weighted departure of the implicit effective wind directions $\theta m_i$ corresponding to the $N_p$ peaks from $\theta$, the mean wind direction over the release period.

At the end of this loop, the optimal estimates of the unknown location ($x_e$, $y_e$) and rate ($Q_e$) of the release are taken as the estimates corresponding to the minimum of the cost function $J$. $J_w$ weights the departure of $\theta m$ from $\theta$ using $\sigma_\theta$, which characterizes here the uncertainty in the effective winds. The misfits between modeled and simulated peak amplitudes (Eq. 4) are not explicitly weighted by the uncertainty in the transport model associated to the comparison between the Gaussian model and instantaneous plume-cross sections or to the configuration of the parameters for this model. However, the direct comparison of $J_w$

and $J_p$ in $J$ implicitly assumes that there is a 100% uncertainty in the skill of the Gaussian model to simulate the amplitude of individual peaks when feeding it with the actual release locations and rates, which is a rather conservative assumption (Ars et al., 2017).

The first results analyzed based on the inversion configuration described above and presented in sections 5.1 and 5.2 have led us to conduct some tests of sensitivity of the inversions: (1) by fixing the location of the source to its actual position and minimizing $J_p$ to get an estimate of the release rates (2) by modifying the formulation of $J_p$ to influence the way it weights the fit to the different peak amplitudes (see section 5.3) and (3) by rescaling $J_w$ to change its relative weight in $J$. Section 5 details these tests and their results. The principle of our method does not apply to releases for which we only have one plume cross-section. In such a case, the amplitude and location of this cross-section do not provide enough information to infer both the source rate and location. Indeed, for any location corresponding to the mean measured wind and thus cancelling $J_w$, the release rate can be fixed to perfectly match the observed plume amplitude and cancel $J_p$. However, the first results analyzed based on the standard inversion configuration described above also showed the limitations of the skill to infer the source location. Therefore, in order to highlight this problem and to strengthen our statistics regarding the skill to infer the release rates, we have included in our analysis the results from a release during which we had one plume cross-section only.

## 4. Data analysis for the configuration of the transport model and of the inversion

### 4.1 Assignment of the background mole fractions

The definition of the background field of $CH_4$ or $CO_2$ for the measurements along the different plume cross-sections can have a strong impact on the derivation of the peak amplitudes. Our modeling framework includes the Gaussian simulation of the plumes from the controlled releases but not a simulation of the background mole fractions over which the plumes represent an excess of $CH_4$ or $CO_2$. We compute a single background value per release. During a given $CH_4$ release, we define the background as the minimum of the corresponding timeseries of measured $CH_4$ mole fractions. Indeed, the variations of $CH_4$ between the peaks that are unambiguously attributed to the plume from the targeted source appear to be quite negligible in most cases, which can be explained by the short duration of the releases. However, the mole fractions were much noisier between the peaks in the $CO_2$ mobile measurement transects, due to potential sources and sinks of $CO_2$ nearby such as vegetation and traffic (e.g. delivery trucks passing frequently along the road surrounding the TADI platform). Therefore, we define the $CO_2$ background value for a given $CO_2$ release as the 5th percentile of the corresponding timeseries of measured $CO_2$ mole fractions. These background values are subtracted from the measurement timeseries for the computation of the observed peak amplitudes.

## 4.2 Configuration of the Gaussian model and identification of the releases for which the modeling framework is suitable

We use the average of the 1-minute data from the Metek 3D sonic anemometer over each release period as inputs to the Gaussian plume model: the average of the standard deviations of velocity fluctuations in the lateral ($\sigma_v$) and the vertical ($\sigma_w$) directions are used to compute the dispersion parameters $\sigma_Y$ and $\sigma_Z$, and the average wind speed $U$ is taken as the effective wind speed $U_e$ driving the Gaussian model.

The inversion method relies on the detection and use of clear peaks in the gas mole fraction timeseries that really correspond to cross-sections from one edge to the other edge of the plumes. Several peaks in the measurements were associated to situations for which the vehicle had to turn (e.g. at the crossing of roads) and thus did not fully cross the plumes. Such peaks are not retained for the inversions. Furthermore, some peaks were measured at locations very far from the area along the road corresponding to the projection of the ATEX zone with the $\theta \pm 2\sigma_\theta$ range of wind directions. The reliability of inversions using such peaks would be very low and we thus exclude all peaks for which the difference between the corresponding $\theta m$ and $\theta$ systematically exceeds 30° whichever location is tested for the source. Due to the complex meteorological conditions during the campaign (60% of the releases were conducted while the wind was lower than 2 ms$^{-1}$) and due to the low number of detected peaks, this selection of the peaks that are suitable for inversion meant that there were not any exploitable peaks for 34 of the controlled releases. Only seven $CH_4$ and nine $CO_2$ releases were thus selected for the inversions (Table 2). This selection of releases slightly narrows the range of release rates tested during the TADI-2018 campaign, but the resulting range (0.3 to 45 gCH$_4$/s and 2 to 150 gCO$_2$/s, see Table 2) still spans three orders of magnitude.

About 30% of these releases were conducted in weak wind speed conditions, with $U < 2$ ms$^{-1}$, which are usually assumed to be challenging for local dispersion modeling (Wilson et al., 1976). Such conditions are associated with complex dispersion patterns of the gases released, and deviate from the validity range of the Gaussian plume dispersion model. We analyze these releases, but our confidence *a priori* in these results was thus weaker than for the other releases and specific statistics are derived in section 5 for cases when $U \geq 2$ ms$^{-1}$.

Table 2 provides information about the release rates, number of peaks, and meteorological parameters for each of the releases to which the inversion was applied. In releases numbers 5 and 6, part of the mole fractions measured in the plume cross-sections (3% and 10% respectively) were above the CRDS analyzer's recommended range for $CH_4$ (above 20 ppm, see section 2.2), with maximum values of ~60 ppm and ~85 ppm, respectively. These are the only releases selected for inversion for which measurements were out of this range. There was only one plume cross-section during release no. 12. Meteorological observations missed for the two last releases (nos. 15 and 16 in Table 2) due to technical problems. For these two releases, meteorological observations from the previous release (i.e. no. 14),

which occurred about nine minutes before, are used for the inversion. For the selected releases which correspond to low wind speed conditions ($U < 2$ ms$^{-1}$), we set a minimum value of 0.3 ms$^{-1}$ for $\sigma_v$, and the effective wind speed of the Gaussian model to $U_e = (U^2 + 2\sigma_v^2)^{1/2}$ (Qian and Venkatram, 2011). Atmospheric stabilities during the selected releases were in the range of neutral to very unstable as all the gas releases were conducted during day time and the observed values of $L$ were negative (Table 2).

Table 2: Releases to which the inversion is applied, with the corresponding release duration, actual release rate ($Q_s$), number of peaks ($N_p$) in the mobile measurement transects, and averaged values of the meteorological and turbulence parameters (mean horizontal wind speed ($U$) and direction ($\theta$), the Obukhov length ($L$), surface friction velocity ($u_*$), and standard deviation of wind velocity fluctuations

($\sigma_u$, $\sigma_v$, and $\sigma_w$)) over the release period.

| Release no. | Gas | Duration (mm:ss) | $N_p$ | $Q_s$ (g/s) | $z_s$ (m) | $U$ (m/s) | $\theta$ (°) | $1/L$ (m$^{-1}$) | $u_*$ (m/s) | $\sigma_u$ (m/s) | $\sigma_v$ (m/s) | $\sigma_w$ (m/s) |
|---|---|---|---|---|---|---|---|---|---|---|---|---|
| 1 | CH$_4$ | 07:48 | 2 | 1 | 2.3 | 2.06 | 294.8 | -0.03 | 0.34 | 0.55 | 0.60 | 0.50 |
| 2 | CH$_4$ | 06:54 | 2 | 0.5 | 2.1 | 2.64 | 290.7 | -0.06 | 0.26 | 0.42 | 0.50 | 0.42 |
| 3 | CH$_4$ | 18 :25 | 6 | 0.3 | 2.1 | 2.86 | 285.7 | -0.08 | 0.23 | 0.48 | 0.41 | 0.42 |
| 4 | CH$_4$ | 08:36 | 4 | 0.5 | 7.0 | 2.90 | 312.6 | -0.02 | 0.31 | 0.49 | 0.50 | 0.42 |
| 5 | CH$_4$ | 08:31 | 4 | 45 | 1.6 | 2.29 | 307.4 | -0.06 | 0.22 | 0.40 | 0.48 | 0.37 |
| 6 | CH$_4$ | 14:25 | 4 | 3 | 1.1 | 1.77 | 156.3 | -0.04 | 0.22 | 0.41 | 0.41 | 0.38 |
| 7 | CH$_4$ | 12:00 | 2 | 0.5 | 2.6 | 2.40 | 142.7 | -0.02 | 0.23 | 0.44 | 0.32 | 0.32 |
| 8 | CO$_2$ | 06:18 | 2 | 150 | 1.6 | 3.32 | 67.42 | -0.01 | 0.37 | 0.67 | 0.58 | 0.48 |
| 9 | CO$_2$ | 08:57 | 2 | 5 | 1.7 | 3.31 | 76.7 | -0.01 | 0.38 | 0.77 | 0.67 | 0.54 |
| 10 | CO$_2$ | 06:39 | 4 | 3 | 0.6 | 2.85 | 55.7 | -0.01 | 0.28 | 0.49 | 0.52 | 0.41 |
| 11 | CO$_2$ | 04:49 | 2 | 2 | 1.9 | 2.19 | 52.1 | -0.01 | 0.25 | 0.39 | 0.44 | 0.35 |
| 12 | CO$_2$ | 04:20 | 1 | 150 | 1.6 | 1.23 | 312.2 | -0.09 | 0.17 | 0.25 | 0.27 | 0.28 |

| 13 | $CO_2$ | 04:30 | 2 | 85 | 1.6 | 1.41 | 304.5 | -0.04 | 0.22 | 0.28 | 0.29 | 0.32 |
| 14 | $CO_2$ | 04:01 | 2 | 60 | 1.6 | 1.26 | 308.1 | -0.16 | 0.19 | 0.34 | 0.31 | 0.28 |
| 15 | $CO_2$ | 04:52 | 2 | 30 | 1.6 | 1.26 | 308.1 | -0.16 | 0.19 | 0.34 | 0.31 | 0.28 |
| 16 | $CO_2$ | 04:00 | 3 | 10 | 1.6 | 1.26 | 308.1 | -0.16 | 0.19 | 0.34 | 0.31 | 0.28 |

## 5. Results

We evaluate the inversion estimates of the rates and locations of the selected releases using the actual values provided by TOTAL. The number of plume cross-sections used by the inversion for individual $CH_4$ or $CO_2$ releases varies from 1 to 6 with a typical range of 2-4 (Table 2).

### 5.1 $CH_4$ releases

Table 3 shows the inverted and actual release rates and location errors for the seven $CH_4$ releases. As an example, the shape of the cost function $J$ and of its components $J_p$, and $J_w$ as a function of the source location within the ATEX zone and the minimum of $J$ are illustrated for release no. 2 in Figure 3 by fixing the release rate to its inversion estimate, and compared to the actual source location (similar figures for all the releases are provided in Figures S2-17 of the SI). This Figure highlights the dominant role of $J_w$ in the determination of the source location. For this release, Figure 4 also shows a comparison between the observed and modeled (using the source location and rate given by the inversion) peaks of $CH_4$ mole fractions for two of the plume cross-sections. For both cross-sections, the maxima of the measurements are larger than that of the modeled gas mole fractions but the modeled plume cross-section is wider, as explained by the use of a Gaussian model which is representative of the average dispersion. However, the modeled and observed integral of the gas mole fractions above the background within the plume cross-sections agree within 26%. The average of this relative difference between the amplitudes of the simulated and observed peaks (comparing the absolute value of the differences to the observed amplitude) over all peaks from all releases is about 43%. The deviation of $\theta m$ from $\theta$ varies from less than 1° to ~27° with average deviation of ~8° over all the peaks in all $CH_4$ releases, while $\sigma_\theta$ varies between 8° and 17°, with an average value of 11°. These values explain that with the inversion estimates of the release location and rate, the value of $J_p$ is smaller than that of $J_w$ (as illustrated in Figure 3).

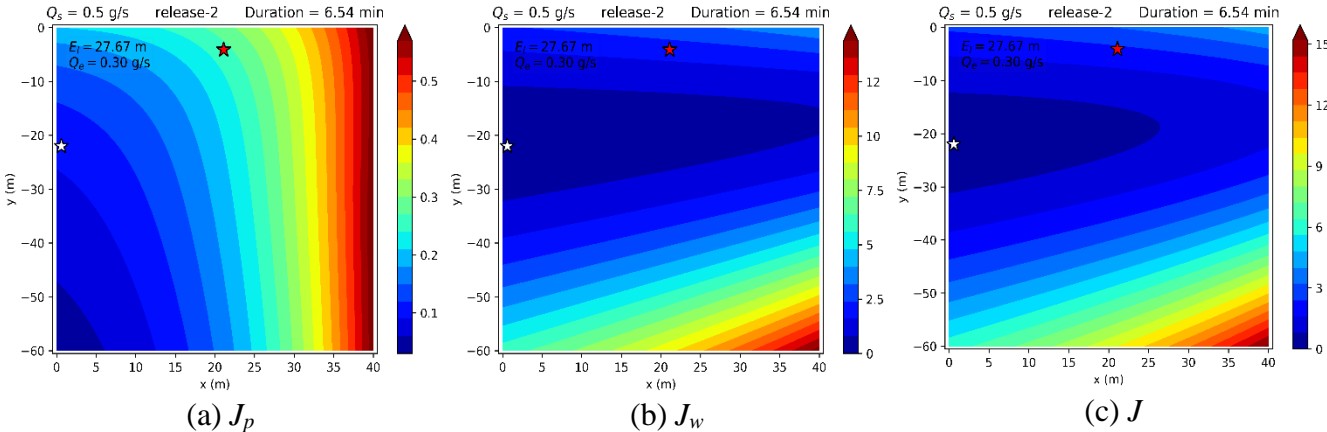

(a) $J_p$        (b) $J_w$        (c) $J$

Figure 3: Contour plots of (a) $J_p$ , (b) $J_w$, and (c) $J$ when fixing the release rate to its inverted value $Q_e$ for release no. 2. Red and white stars respectively show the actual and inverted source locations.

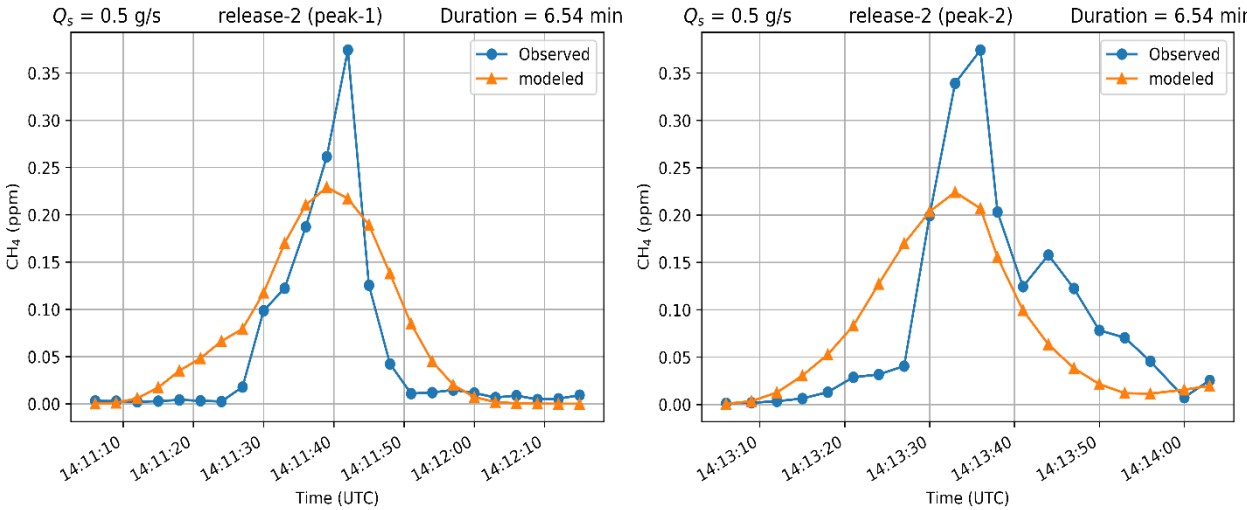

Figure 4: Observed and modeled peaks in the $CH_4$ mole fractions for two plume cross-sections used in the inversion for release no. 2, using the estimates of the source location and rate from the inversion.

For each controlled release, the error in the estimate of the source location (the "location error" hereafter) $E_l$ is defined by the Euclidean distance between the inverted and actual source. It varies from 8.1 m to 53.9 m, with an average value of 28.6 m across all the selected $CH_4$ releases (Table 3). Figure 5(a) shows a comparison between the estimated and actual release rates for these releases. The relative estimation

error for the release rates (dividing the absolute value of the estimation error by the actual emission rate) varies from less than 10% (for release no. 4) to ~82% (for release no. 5) (Table 3, Figure 5(a)). These results indicate that the inversions lead to an average relative error of 31.2% in the release rate estimates. In most of the cases, the estimates of the rates are within a factor of two from the actual ones, except for release no. 5, for which the actual release rate is underestimated by a factor of ~5.5. The underestimation of the rate for release no. 6 is the second-worst case with ~50% relative error. The small percentage of mole fractions measured above the analyser's operational range for $CH_4$ during releases nos. 5 and 6 (section 4.2) does not sufficiently explain why these releases correspond to the poorest results. Selecting the cases for which $U \geq 2$ ms$^{-1}$ slightly decreases the average relative error to 28%, release no. 6 being the only one for which $U < 2$ ms$^{-1}$. However, ignoring the results for the worst case (release no. 5), the average relative error in the release rate is ~23%. In most of the cases, the actual release rates are underestimated by the inversion (release nos. 4 and 7 being exceptions).

Table 3: Summary of the results from the inversions with comparisons between the actual and inverted source locations and rates for the $CH_4$ releases.

| Release no. | Gas | $Q_s$ (g/s) | Inversions minimizing $J$ (Eq. (3)) | | | | | Inversions minimizing $J^{log}$ (Eq. (7)) | | | | |
|---|---|---|---|---|---|---|---|---|---|---|---|---|
| | | | Source fixed to its actual location | | Deriving both the rate and location of the source | | | Source fixed to its actual location | | Deriving both the rate and location of the source | | |
| | | | $Q_e$ (g/s) | Rel. error | $Q_e$ (g/s) | Rel. error | $E_l$ (m) | $Q_e$ (g/s) | Rel. error | $Q_e$ (g/s) | Rel. error | $E_l$ (m) |
| 1 | CH$_4$ | 1 | 0.55 | 45.0% | 0.80 | 20.0% | 26.8 | 0.70 | 30.0% | 1.05 | 5.0% | 26.8 |
| 2 | CH$_4$ | 0.5 | 0.25 | 50.0% | 0.30 | 40.0% | 27.7 | 0.25 | 50.0% | 0.30 | 40.0% | 27.7 |
| 3 | CH$_4$ | 0.3 | 0.20 | 33.3% | 0.25 | 16.7% | 21.5 | 0.20 | 33.3% | 0.25 | 16.7% | 21.5 |
| 4 | CH$_4$ | 0.5 | 0.50 | 0.0% | 0.50 | 0.0% | 8.1 | 0.55 | 10.0% | 0.60 | 20.0% | 7.7 |
| 5 | CH$_4$ | 45 | 6.55 | 85.4% | 8.05 | 82.1% | 38.8 | 7.55 | 83.2% | 9.05 | 79.9% | 38.8 |
| 6 | CH$_4$ | 3 | 0.70 | 76.7% | 1.50 | 50.0% | 53.9 | 1.50 | 50.0% | 3.00 | 0.0% | 53.9 |
| 7 | CH$_4$ | 0.5 | 0.40 | 20.0% | 0.55 | 10.0% | 23.2 | 0.55 | 10.0% | 0.75 | 50.0% | 23.2 |

## 5.2 CO$_2$ releases

The general patterns and relative weight of $J_w$ and $J_p$ for the $CO_2$ releases is similar to that for the $CH_4$ releases. The average relative difference between modeled and observed peak amplitudes is about 32%. The deviation of $\theta m$ from $\theta$ varies from less than 1° to ~26° with an average value of ~7° over all the

peaks in all $CO_2$ releases, while $\sigma_\theta$ varies from 10° to 14° with an average value of 12°. Again, this is associated with lower values for $J_p$ than $J_w$ (not shown).

Table 4 and Figure 5(b) compare the estimates of the $CO_2$ release rates and locations to their actual values. The location error is, on average, ~36 m. For all the nine $CO_2$ releases that have been analyzed, the emissions are estimated within a factor of 1.4 of the actual emissions. The relative error in the release rate

estimates varies from less than 2% (release no. 10) to 28.6% (release no. 8), and on average is 17.2%. Ignoring the five releases corresponding to $U < 2$ ms$^{-1}$, the average relative error for the estimates of release rates significantly decreases to 11.6%. Errors on the estimates of the rate and location for release no. 12, during which we have one plume cross-section only, are close to the average errors. This highlights the limitation of the skill to provide a precise estimate for the release location whatever the number of

plume cross-sections used. As was observed for the $CH_4$ releases, there is a general tendency of the inversions to underestimate the actual $CO_2$ release rates (with two exceptions: release no. 10 and 12).

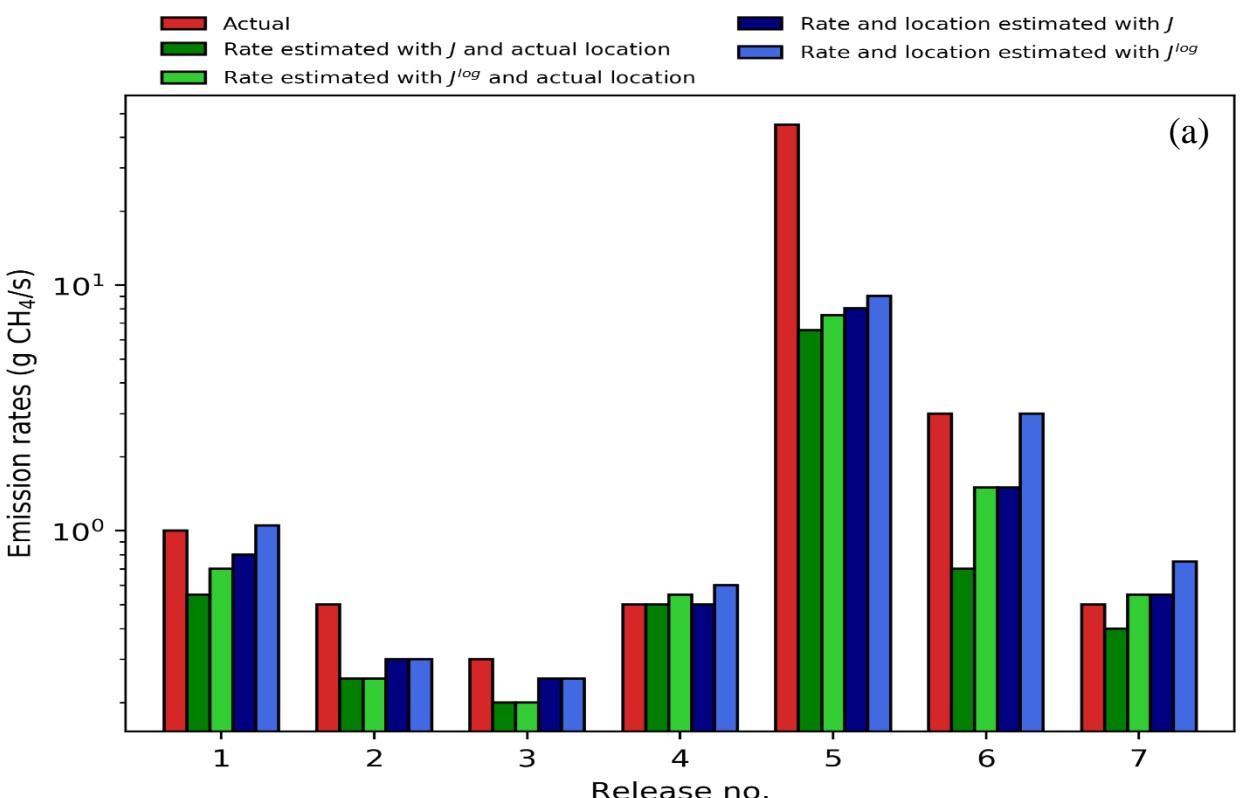

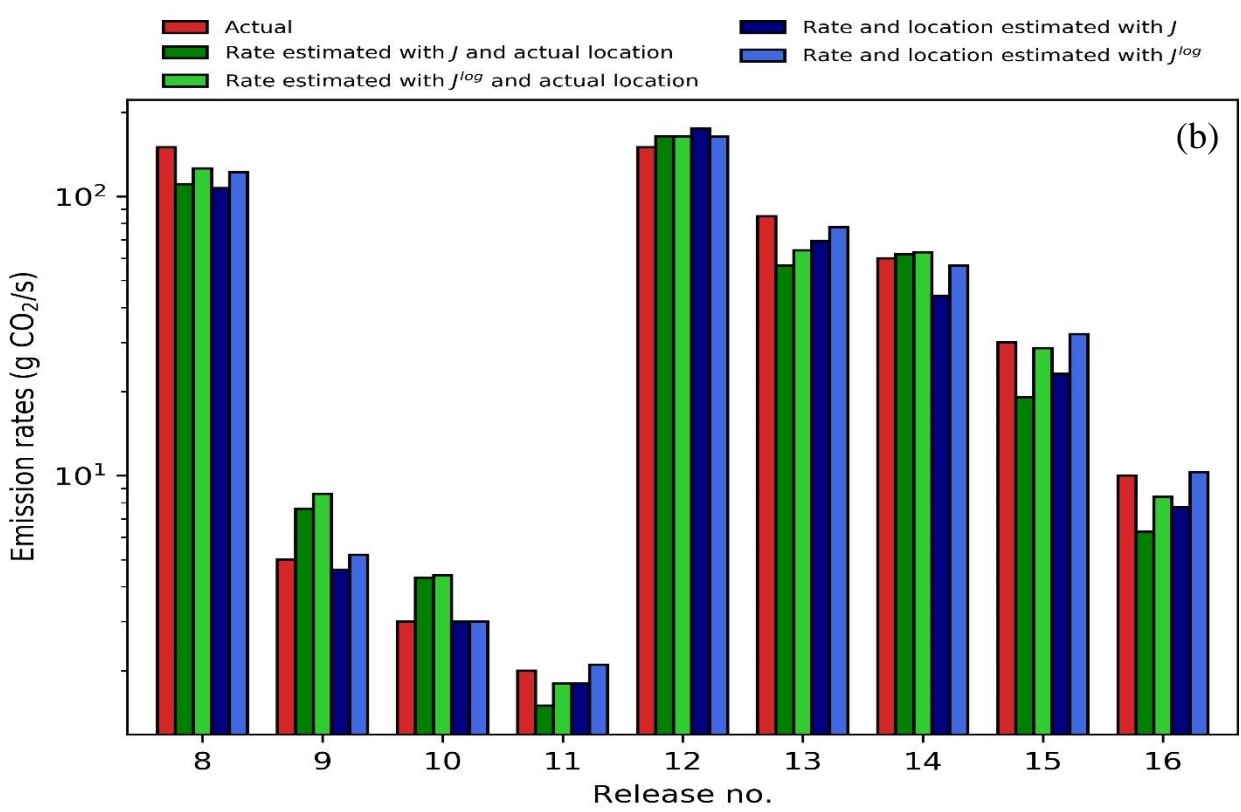

Figure 5: Comparison of the estimated and actual emissions rates of the (a) $CH_4$ and (b) $CO_2$ releases.

Table 4: Summary of results from the inversion with comparisons between the actual and inverted source location and rates for $CO_2$ releases.

| Release no. | Gas | $Q_s$ (g/s) | Inversions minimizing $J$ (Eq. (3)) | | | | | | Inversions minimizing $J^{log}$ (Eq. (7)) | | | | | |
| --- | --- | --- | --- | --- | --- | --- | --- | --- | --- | --- | --- | --- | --- | --- |
| | | | Source fixed to its actual location | | Deriving both the rate and location of the source | | | | Source fixed to its actual location | | Deriving both the rate and location of the source | | | |
| | | | $Q_e$ (g/s) | Rel. error | $Q_e$ (g/s) | Rel. error | $E_l$ (m) | | $Q_e$ (g/s) | Rel. error | $Q_e$ (g/s) | Rel. error | $E_l$ (m) | |
| 8 | $CO_2$ | 150 | 110.6 | 26.3% | 107.1 | 28.6% | 21.5 | | 126.1 | 15.9% | 122.1 | 18.6% | 21.5 | |
| 9 | $CO_2$ | 5 | 7.6 | 52.0% | 4.6 | 8.0% | 43.9 | | 8.6 | 72.0% | 5.2 | 4.0% | 43.9 | |
| 10 | $CO_2$ | 3 | 4.3 | 43.3% | 3.0 | 0.0% | 32.3 | | 4.4 | 46.7% | 3.0 | 0.0% | 33.2 | |
| 11 | $CO_2$ | 2 | 1.5 | 25.0% | 1.8 | 10.0% | 25.0 | | 1.8 | 10.0% | 2.1 | 5.0% | 25.0 | |
| 12 | $CO_2$ | 150 | 164.1 | 9.4% | 175.1 | 16.7% | 26.1 | | 164.1 | 9.4% | 163.6 | 9.1% | 23.3 | |
| 13 | $CO_2$ | 85 | 56.6 | 33.4% | 69.1 | 18.7% | 44.8 | | 64.1 | 24.6% | 77.6 | 8.7% | 44.8 | |

| 14 | $CO_2$ | 60 | 62.1 | 3.5% | 44.1 | 26.5% | 44.8 | 63.1 | 5.2% | 56.6 | 5.7% | 44.8 |
| 15 | $CO_2$ | 30 | 19.1 | 36.3% | 23.1 | 23.0% | 44.8 | 28.6 | 4.7% | 32.1 | 7.0% | 44.8 |
| 16 | $CO_2$ | 10 | 6.3 | 37.0% | 7.7 | 23.0% | 37.9 | 8.4 | 16.0% | 10.3 | 3.0% | 37.9 |

## 5.3 Understanding biases in the release rate and location estimates: sensitivity tests

### 5.3.1 Biases and results when using fixing the source to its actual location.

The results for both $CH_4$ and $CO_2$ releases indicate that for ~90% of the cases, the release rates are underestimated by the inversion. However, the locations of the sources are generally found to be too far from the main measurement transects compared to their actual position, an inversion bias which should rather lead to an overestimation of the release rates. Experiments using the same inversion framework but fixing the source location to its actual position (minimizing $J_p$) leads to a ~44% and ~30% average relative
error in the estimate of the $CH_4$ and $CO_2$ release rates respectively, i.e. to larger errors (see the detailed results in Tables 3 and 4).

Actually, the underestimation of the release rates when fixing or deriving the release location coincides with the underestimation of most of the peak amplitudes. Across the different peaks corresponding to a given release, the relative difference between the amplitudes of the simulated and observed peaks is highly
variable and it appears that the system is often highly sensitive to one or two peaks for which it provides a slight overestimation, balanced by a large underestimation of the other peaks. This phenomenon appears to be have an origin that could also explain the limited skill for deriving precise estimates of the release locations. Indeed, a potential explanation for the overestimation of the distance to the source and for the underestimation of the release rates is thus that the term $J_p$ of the cost function does not force enough the
results to correspond to the source location and rate that provide a good fit to most of the peak amplitudes. In particular, it does not force enough the results to correspond to the right variations in terms of peak amplitude from one plume cross-section to the other. With such a lack of constraint regarding the relative amplitude of the different peaks, the potential to find the actual release location is strongly limited, and with values for $J_p$ much lower than those for $J_w$, a primary driver of the minimization of $J$ is that of $J_w$ by
localizing the source as far as possible.

### 5.3.2 Least-squares fitting of the order of magnitude of the peak amplitudes rather than of the values of these amplitudes

Therefore, a sensitivity test is performed to put more emphasis on a better fit to the different peak amplitudes and to loosen the strongest constraints towards specific peaks. The term $J_p$ is modified to
weight the misfits between the modeled and measured amplitudes of the plume cross-sections in terms of order of magnitude using a logarithmic scale:

$$J_p^{log} = \sum_{i=1}^{N_p} \left[ \frac{log(1+Ao_i) - log(1+Am_i)}{log(1+Ao_i)} \right]^2 \qquad (6)$$

In a new series of estimations, the inversion minimizes

$$J^{log} = J_p^{log} + J_w \qquad (7)$$

instead of $J$. The corresponding results (Tables 3 & 4 and Figure 5) are slightly better than that obtained when minimizing $J$.

Minimizing $J^{log}$ for the $CH_4$ releases, the location errors vary from 7.7 m to 53.9 m, with an average value of 28.50 m (Table 3) and the relative error in the estimates of the release rates vary from ~5% (release no. 1) to ~80% (release no. 14), with a ~30% average value. These scores are very similar to that when
minimizing $J$. Minimizing $J^{log}$ for the $CO_2$ releases, the average location error is 35.4 m, which, again, is similar to the average location error when minimizing $J$. However, there is a significant improvement in the estimate of the $CO_2$ release rates when minimizing $J^{log}$: the relative error in this estimate varies from less than 2% to 18.6%, with an average relative error of 6.8%. For all the nine $CO_2$ releases, minimizing $J^{log}$ leads to release rate estimates within a factor of 1.2 of the actual release rates.

A more general improvement when minimizing $J^{log}$ is that there is no general tendency to underestimate the release rates, with now 60% of cases for which the release rate is actually overestimated. However, the tendency to overestimate the distance of the source from the main mobile measurements transects persists: $J^{log}$ is dominated by $J_w$ such as $J$, and the capability to localize the sources keeps on being limited. This reveals a persistent tendency of the system to underestimate release rates. However, this tendency is
now balanced by the system's opposing tendency to increase the release rates to compensate for the distance between the source and the plume cross-sections being overestimated. Indeed, when the source location is fixed to its actual position, the minimization of $J_p^{log}$ (like the minimization of $J_p$), tends to underestimate the release rates and yields a ~38% and ~23% relative error in the estimate of the $CH_4$ and $CO_2$ release rates, respectively.

**5.3.3 Sensitivity to the assessment of the model error**

A last series of sensitivity tests where both the release location and rates are derived is performed by varying the relative weight of $J_w$ in $J$ and $J^{log}$. The aim is to generate situations for which this relative weight of $J_w$ is comparable to that of $J_p$ and $J_p^{log}$ and thus to improve the estimate of the release location. In these tests, the cost functions are rewritten $J = J_p + \lambda J_w$ and $J^{log} = J_p^{log} + \lambda J_w$ which is implicitly
equivalent to assume that we have a relative error of $\sqrt{\lambda}$ when modeling the plume areas $Am_i$ in $J_p$ or when modeling the $log(1 + Am_i)$ in $J_p^{log}$.

When $\lambda = 0$, $J = J_p$ and $J^{log} = J_p^{log}$ and the location is constrained by the varying amplitude of the different peaks only. Figure 3 and Figures S2-S17 of the SI show that $J_p$ is smooth and nearly systematically reaches its minimum at a border of the ATEX zone (which is generally different from that corresponding to the minimum of $J_w$). The situation is similar for $J_p^{log}$. This generally yields location errors (~42 m average location error with both $J = J_p$ and $J = J_p^{log}$ for $CH_4$, ~37 m and ~38 m average location errors respectively with $J = J_p$ and $J = J_p^{log}$ for $CO_2$) that are larger than that when using $\lambda = 1$ (~29 m and ~36 m average location errors for $CH_4$ and $CO_2$, respectively for both $J$ and $J^{log}$ with $\lambda = 1$) (see Figures S18 and S19 of the SI). One explanation is the lack of plume transects to provide a precise constraint on the source location. For illustration purpose, one can see that an infinity of locations correspond to the relative amplitude of two plume transects. One of the role of $J_w$ is actually to complement this source of information on the source location.

The analysis of the average location errors as a function of $\lambda$ (Figures S18 and S19 of the SI) shows that for the $CH_4$ releases (when minimizing either $J$ or $J^{log}$) the optimal value of $\lambda$ for localizing the source would be $\lambda = 1$, i.e. using the default inversion configurations. For the $CO_2$ releases, this optimal value becomes 0.004 (i.e. a relative model error of ~6%) when minimizing $J^{log}$, and 0.016 (i.e. a relative model error of ~13%) when minimizing $J$. Furthermore, the curves of the average location errors as a function of $\lambda$ for $CH_4$ releases when minimizing either $J$ or $J^{log}$ have local minima with values close the optimal one obtained for $\lambda = 1$ (for $\lambda = 0.016$, i.e. a relative model error of ~13%, for $J$ and 0.008, i.e. a relative model error of ~9% for $J^{log}$). With such values for $\lambda$, some of the releases are located well inside the ATEX zone (see Figures S20 and S21 of the SI). However, such levels of model error are probably highly optimistic for the Gaussian model, and most of the releases keep on being located on a boundary of the ATEX zone since the resulting $J$ or $J^{log}$ functions of the release location keep on being quite smooth. In all cases, the location error remains quite high and the estimate of the release rate is generally the closest to the actual rate for $\lambda$ close or equal to 1. These results support the assumption that the lack of plume transects (and even more of plume transects which have a significant weight in the minimization of $J_p$ or $J_p^{log}$) coupled to the model error probably explains the limitation of the skill to localize the source.

## 6. Discussion

We developed an inversion framework which does not derive explicit estimates of the uncertainties associated to its release rate and location estimates (unlike statistical frameworks such as that of Ars et al., 2017). We did not attempt at conducting sensitivity or ensemble computations to derive such theoretical uncertainties and rather entirely relied on comparison to the actual release rates and locations to assess the precision of our inversions in an objective way. Our inversion system provided estimates of the $CH_4$ and $CO_2$ release rates with ~10%-40% average relative errors (depending on the inversion configuration or on the series of tests) over the wide range of rates tested during the TADI campaign. The more complex background conditions during the $CO_2$ releases did not appear to be a limitation for the

inversion which provided more precise estimates of the $CO_2$ release rates than of the $CH_4$ release rates on average. The $CO_2$ and $CH_4$ measurement precision is very good and the impact of the measurement errors is negligible in our computations. In such conditions, the linearity of the local scale dispersion of $CO_2$ and $CH_4$ prevents from assuming that the model and the inversion can behave better for $CO_2$ releases than for $CH_4$ releases. Therefore, this difference of average release rate precision can be attributed to the changes in term of meteorological conditions between the $CH_4$ releases and the $CO_2$ releases. These conditions appear to be an important driver of the release rate inversion precision. Even though the estimates for low wind speed were not associated with much larger estimation error, the specific variations of the wind for each release appear to play a critical role in the ability to fit the various amplitudes of the plume cross-sections. The particularly challenging meteorological conditions encountered during the campaign probably played a critical role in the limitation of the ability of the inversion to retrieve the location of the releases. The system achieved a ~30-40 m precision for such an estimation with mole fractions measured at ~50-150 m from the source most of the time. Such an error is quite large when compared to the dimension of the ATEX zone.

However, our results in terms of release rate estimates and thus our inversion approach appear to be promising given the very complex conditions of the campaign with very brief releases and low but highly varying wind conditions. ~10%-40% precision estimates for the release rates can be very useful to assess the level of emissions from industrial sites (Brantley et al., 2014). Previous studies dedicated to the estimate of release rates from point sources using mobile measurements across the plumes and atmospheric dispersion models (such as Brantley et al., 2014; Foster-Wittig et al., 2015; Albertson et al., 2016) documented similar typical average precisions of ~20-30% but they relied on releases and measurement timeseries lasting at least 20 minutes. Longer release durations (e.g. at least 30 minutes) would enable a much higher number of plume cross-sections to be measured around the site and this could ensure much more favorable inversion conditions. Caulton et al. (2018) recommended to use at least 10 plume cross-sections to reliably constrain atmospheric variability and reduce the uncertainties in the estimation of the emission rates using mobile measurements. However, our results demonstrate that we can achieve a good estimation precision with a much smaller number of plume cross-sections.

Some major improvements can be foreseen to strengthen the measurement and inversion framework. The general tendency of the atmospheric transport and inversion framework to underestimate the release rates (compensated by its tendency to overestimate the distance between the source and the plume cross-sections when deriving the release locations together with the release rates using a logarithmic cost function) can actually be related to the effective release injection height (Yacovitch et al., 2020). In the inversion computations, this height is fixed to the actual source height $z_s$. However, the gas is released with significant velocity and difference of temperature relative to the ambient environment, leading to some important rising of the plume to several meters above the actual release point. Images taken with hyperspectral cameras by other participants in the TADI campaign during some of the releases indicated

that the released plume had significant momentum which caused it to rise by approximately 2-3 m (likely
up to 10 m for some releases) above the actual release points. An estimate of the effective injection height
$z_e$ accounting for plume rise (Briggs, 1975) may thus have to be considered in the model. In principle, the
inversion could optimize the injection height estimate $z_e$ as well as the release location and rate. However,
the problem would be too underconstrained for the TADI campaigns given the limited number of plume
cross-sections for each release, and thus, because of the brevity of the release. Some sensitivity tests (not
shown) were conducted by increasing incrementally the release height $z_e$ in tests identical to those
presented in section 5. The results show that such an increase can rapidly (after the addition of few meters
to $z_s$) yield release rate estimates that are larger than the actual rates. Precise estimates of the injection
height are thus required to ensure an improvement of the results presented here.

Uncertainties in the atmospheric stability and other meteorological and turbulence parameters can be a
critical source of errors, especially when targeting short releases. Here, the parameterization of the
Gaussian model relied on meteorological and turbulence measurements that may be poorly representative
of the atmospheric conditions between the location of the release and the plume measurement cross-
sections for some releases. Using the integrals of the gas mole fractions within the plume cross-sections
as observations limits the impact of uncertainties in the horizontal diffusion. However, the vertical
dispersion is generally more important than the horizontal dispersion and uncertainties in vertical
dispersion can significantly impact the inversion of the release rate (Caulton et al., 2018). The strong
underestimation of the $CH_4$ emission in release no. 5 is probably due to a poor representation of the
atmospheric stability conditions. Mobile measurements taken at different heights simultaneously could
help overcome such an issue as well as that of the derivation of the release injection height.

A result from the current shortcomings when applying our inversion technique to the practical test cases
presented here is the limited ability to extract information on the source location, or to derive precise
estimates for both the locations and rates of the releases, even when exploiting the information from more
than four plume cross-sections. We showed that this limitation is partly connected to the lack of weight
of $J_p$ in our total cost functions in practice but also to the lack of weight of many of the plume cross-
sections in $J_p$ itself. The sources of model errors highlighted above explain it for a large part. However, a
better assessment of the model errors as a function of the plume transect without using the knowledge on
the actual source rate and location (potentially with the kind of techniques envisaged in Ars et al., 2017)
could help refine the definition of $J_p$. The conservative assumption regarding this error that is implicitly
made in Eq (4) partly explains that $J$ is dominated by $J_w$ and thus the lack of fit to the different plume
cross-sections during a given release, but the crude reweighting of $J_w$ does not solve for the overall
problem of the source location. $J_p$ should balance misfits to the observation with a model error that is
consistent with such misfits. More sensible estimations of the ability of the model to simulate the
amplitude of the different peaks lower than 100% could be used to increase the weight of individual
departures from the observed amplitudes.

As mentioned earlier, many of the releases during the TADI campaign were conducted under weak wind conditions. The Gaussian plume models have limited applicability in such weak wind conditions (Thomson and Manning, 2000) even though they are shown to provide reasonable dispersion simulations under moderate to strong wind conditions. For practical reasons, the selection of the Gaussian model, which is fast and relatively easy to implement and control, appeared to be optimal for the initial tests of the inversion framework and the simulation of plumes for a very wide range of potential source locations in the inversion scheme. However, in principle, more advanced models like Lagrangian dispersion models and/or Computational Fluid Dynamics (CFD) models are more suitable for atmospheric dispersion in such extreme meteorological conditions (Tominaga and Stathopoulos, 2013). Combining such models with our inversion approach could provide opportunities to account for the variations of the wind in space and time and for vertical profiles of the releases. CFD models like Large Eddy Simulations (LES) models simulating instantaneous plumes and in particular the turbulence could also allow to investigate the width of instantaneous plume cross-sections, which could add some significant constraints for the unambiguous estimate of both the rate and location of the releases. However, exploiting these potential assets of such models is challenging in practice, and due to their computational cost, they may be difficult to use for the inversion of the source location. A hybrid approach combining Gaussian models and more complex ones for the joint inversion of the source location and rate might be a solution to this problem.

## 7. Conclusions

In this study, a simple atmospheric inversion modeling framework was developed for the localization and quantification of unknown $CH_4$ and $CO_2$ releases from point sources based on near-surface mobile gas mole fraction measurements. The inversion framework relies on a local-scale Gaussian plume dispersion model and it exploits the position and amplitude of the different gas mole fraction plume cross-sections to infer the source locations and rates. We used it to analyze a series of experiments with very brief controlled releases of $CH_4$ and $CO_2$ covering a wide range of release rates during the TADI-2018 campaign. These releases were detected and quantified using a series of mobile measurement transects across the corresponding plumes made with instruments onboard a car that drove along roads around the emission area. Results indicate a ~10-40% average error on the estimate of the release rates, and ~30-40m average errors in the estimates of the release locations. Considering the challenging atmospheric transport and emission conditions during the TADI-2018 campaign, the limited number of plume cross-sections (typically 2-4) per release, and the limitations of the Gaussian dispersion modeling framework to simulate instantaneous plume cross-sections for short durations, the good inversion results in terms of rates for both $CH_4$ and $CO_2$ releases appear to be encouraging. However, some methodological developments seem to be required to improve the robustness of the estimates for the release locations.

**Code and data availability.**

The data and code are accessible upon request from the corresponding author
(pramod.kumar@lsce.ipsl.fr).

**Author contributions.**

PK implemented the inverse modeling system and performed the model simulations and inversions. GBr
and PK designed the inverse modelling system and "LSCE people & IFPEN" participated to the design
of the observation strategy. CYK, OL, SG, GBe, FM conducted the measurement campaign and processed
the data together with PK. PK and GBr prepared and reviewed the paper with critical contributions from
all co-authors. All co-authors participated to the discussions regarding the results of the experiments.

**Competing interests.**

The authors declare that they have no conflict of interest.

**Acknowledgements.**

We wish to thank the reviewers of this article, in particular Joseph Pitt, for their very useful comments.

**Financial support.**

This research has been supported by Chaire Industrielle TRACE (grant no. ANR-17-CHIN-0004-01),
which is cofunded by the French National Research Agency (ANR), TOTAL Raffinage Chimie, SUEZ,
and Thales Alenia Space (TAS).

**Review statement.**

This paper was edited by Huilin Chen and reviewed by Joseph Pitt and one anonymous referee.

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
