# Peer review of "Mobile atmospheric measurements and local-scale inverse estimation of the location and rates of brief CH4 and CO2 releases from point sources"

_Atmospheric Measurement Techniques, 2020_

## Referee Comment (RC1) · Anonymous Referee #1 · 3 Sep 2020

Summary:

The authors take advantage of the TOTAL Anomaly Detection Initiative platform in the south of France to the skill of an inversion framework that uses in-situ measurements from a mobile advice and a Gaussian plume model. The authors attempt to identify the location and magnitudes of sixteen controlled releases of CO2 and CH4 from a platform that is able to reproduce many common release scenarios that one might encounter on operational sites. The authors contest that their inversion system has some skill – especially in terms of estimating the magnitude of the leak. The main source of errors

is attributed to atmospheric turbulence.

The paper covers a topic relevant for AMT. It is well structured. I recommend publication after the comments listed below are considered by the authors.

Although well structured, many details of the paper were hard to follow. The clarity of the paper (e.g. wording, etc.) could be improved and the manuscript would benefit from a good editorial review. I did not include these types of suggestions in this review.

Comments:

(1) Can the authors note on Figure 1 or somewhere else where the "true" releases were from (aka which star)? I notice that one release site is much further than the others. Is there anything about this location that follows through to the results (e.g. harder to pinpoint or quantify?).

(2) Can the authors further clarify why the Gaussian Plume model is a reasonable transport model for their study given some of the inherent drawback the authors note in paragraph 255. I found this discussion difficult to understand. Beyond the points that the authors list, isn't this application based on how field data is collected – given that the study has constant emissions for each release, are the measurement averaging time comparable to the source-to-receptor travel time? It seems like most of their reasons have to do with the fact that the alternatives are just to difficult to implement. How do expect the Gaussian plume model assumption to impact the uncertainties on their results?

(3) Can the authors better explain how the authors used the Gaussian model to simulate mole-fractions (Am) and then use within the inversion to minimize the cost function? Where is xe, ye, and Qe specified in the equations? It seems like the authors varied the release rate (Qs) which I assume goes into the Gaussian model to create an ensemble of modelled concentrations (Am)? Then is ye is just the length of the centroid to the location of the observed mole fraction that with that minimizes the sum

of the squared residuals for the two terms in equation 3? Same with Qe? What type of minimization scheme did the authors use? These sections need a lot more clarify. For example, better nomenclature would help explain how the authors run their inversion with the plume model. A simple flow diagram would help here too.

(4) Similar questions include: How many ensembles do the authors have for each Ao for a given grid locations? Their grid is 2,400 – I assume the authors used all of the Ao one-minute data within the "peaks" (obviously excluding those due to turns, weak winds, other criteria mentioned) or did the authors average in some way? How many Ao measurements did this amount to for each release? Are the number of Ao the same for both gases? All these points should be better clarified in the draft because it is very hard for the reader to follow the method, how it was applied, and thus be able to understand the results and discussions.

(5) It seems like one of the major assumptions of the work is that the authors provide the actual source height of each release to the inversion system. Later, the authors explain that this height might be artificially too low due to turbulence. I would recommend running a sensitivity test varying the source height. For one, it might show if the source height is effectively "higher", then it explains their results. It would also show the sensitivity of the results due to this assumption.

(6) It is unclear how/if the authors estimate uncertainties on their estimated parameters. Can the authors please provide more information? If the authors did not estimate them, I recommend a sensitivity analysis as noted above to help provide some measure of the variability of their estimates to assumptions. The authors must make some attempt at estimating uncertainties.

(7) Can the authors provide a histogram (or something similar) to show how the 5th percentile value compares to the other Ao mole fractions for justification of background? I would expect that the surrounding vegetation that the authors mention doesn't have too much impact given the short duration of their measurement time period but trucks

passing by would. If these are just spikes in the data, I can imagine that the authors could just remove them.

(8) The location error for CH4 is very large especially given the size of the domain and potential release location sites (in general – how far apart are these? Hard to tell from the schematic – I would assume that the average spacing between these would act as a design requirement for the inversion). I suspect that this may have something to do with the fact that the inversion is very underdetermined. Is there any additional information that the authors could provide the system to help reduce the location errors? For example, the authors noted the potential release sites – if these are known a priori, the authors could limit their estimated locations to solely these grid cells. At the very least, the authors could probably make some gross a priori assumptions about where they are not release locations to eliminate part of the solution space.

(9) Following (8) and (1) – can the authors say something about the location of the "potential" release points (some are bunched together while another is much further away and their results?

(10) I am not sure if averaging such a large variation of errors is really a representation of the expected errors – especially for only 7 samples. Can you justify?

(11) Given the range of the release magnitudes – I think it would be nice to see some standard error statistics instead of relative error in the results.

(12) Why aren't there error bars on Figure 5? Having CH4 stacked on CO2 begs for a comparison between the two but they aren't because releases go from 1-16. I would recommend putting these plots side by side instead.

(13) Is one of the main reasons the authors used CO2 and CH4 is because of the background issue? For replication sake, it would have been nice if the authors mimicked their release of CO2 and CH4 (aka same locations, magnitudes, durations, etc.) to be able to compare these. I understand that the authors cannot rerun the experiment but

maybe the authors can group the releases into "similar" types between the two gases to help with the interpretation of results later on?

(14) What can the authors say about the practical application of their results? If the authors need 30 min of sampling – e.g. to improve their estimations – is this typical duration of an intermittent event? How many samples would the authors be able to get in a realistic event? The authors will always have a sampling bias during the day so what does say about their methods? How would the authors extend this for something more useful and practical for operators?

(15) Again, I think presenting the results in relative errors is a bit misleading and I don't think that the authors demonstrated much skill in actually identifying the location of the leak – which is key for the application of this work. I don't think this warrants a rejection of the paper but a more realistic description of their results and methods. I think that detecting and quantifying release may just require a better transport model – even if it is just slightly more sophisticated (e.g. AEROMOD).

(16) The authors suggestion of a hybrid approach is intriguing. Is there any other work that explored these ideas?

---

## Referee Comment (RC2) · Joseph Pitt (Referee) · 11 Nov 2020

This study presents a new approach to determine the location and rate of point source emissions, and tests the method using a series of controlled releases. Mobile measurements of mole fraction are made on board a car, which performs repeated transects through the plume downwind of the release. A Gaussian plume model (driven by local meteorological measurements) is used to simulate mole fractions at the measurement locations for an ensemble of release locations and emission rates. The offset between modelled and measured locations of the plume centre is calculated, in addition to the difference between the integrated plume enhancement (a.k.a. plume amplitude) for the modelled and measured datasets. Estimates for both release location and emission rate are derived by minimising a cost function that seeks to reduce these two measures of model-measurement mismatch.

The study is well-motivated and the details of the experiment are clearly described. Unfortunately, if I've understood it correctly, I think there is a fundamental problem with the method developed here. As far as I can tell, there is insufficient information content in the plume amplitude and plume-centre location to constrain both the location and rate of emissions. It seems like the plume-centre location can be used to constrain the location of the release to a line along the average wind vector, while the plume amplitude can either constrain the emission rate for a given release location on this line, or the release location for a given emission rate. Take the following example where the wind is perpendicular to the transects:

[Figure]

In this case any release location along the dotted line will result in the same modelled plume-centre location. But a source at release point 2 with a low emission rate will produce the same plume amplitude as a source at release point 1 with a high emission rate. These would produce the same value of J for this transect.

In all but one case presented in this study there are multiple transects of the plume. This adds extra information to the case above. But I don't think it is being used to constrain the location in a useful way. Because the emission rate $Q_e$ does not impact $J_W$, J is minimised by setting the location of the release to minimise $J_W$, then setting the release rate to minimise $J_p$ given this location. If the plume centre-location is different for two transects then $J_W$ is minimised by moving the source location further away from the

transects. Figure 3 is the perfect example of this in action. $J_W$ alone sets the source location, but the x-value of this source location is purely an artefact of the way the cost function has been constructed.

In section 5.3 it is stated that $J_p$ does not "push far enough for finding a source location". But in most cases it has no impact on the estimated source location at all, I suspect for the reasons outlined above. This is apparent from tables 3 and 5 – the locations are usually the same regardless of whether $J$ or $J^{log}$ is used as the cost function, because in both cases $J_W$ is the same. In cases where there is a difference I guess that it probably arises from some combination of the geometry of the ATEX zone boundary and the discretisation of release locations and emission rates.

It's entirely possible that I've misunderstood what's going on here – if so then I'm sure the authors can put me straight! But until I have faith in the overall approach, I can't recommend that this paper is published. If the authors can convince me that the method is sound then I'm happy to provide more detailed feedback on specific points. Otherwise I think the best option might be to reject this paper in its current form and consider what additional information could be used to better constrain the problem. One obvious candidate would be to use the plume width in some capacity, but I think that to do so would require a more complex model, as one would need to simulate the likely width of the instantaneous plume (rather than the time-averaged plume represented in the Gaussian plume model). Perhaps that is a bad idea… but either way I think some additional constraint is required in order to render this approach useful in determining source location as well as emission rate.

---

## Author Comment (AC1) · 20 Jan 2021

**GENERAL COMMENTS: The authors take advantage of the TOTAL Anomaly Detection Initiative platform in the south of France to the skill of an inversion framework that uses in-situ measurements from a mobile advice and a Gaussian plume model. The authors attempt to identify the location and magnitudes of sixteen controlled releases of CO2 and CH4 from a platform that is able to reproduce many common release scenarios that one might encounter on operational sites. The authors contest that their inversion system has some skill-especially in terms of estimating the magnitude of the leak. The main source of errors is attributed to atmospheric turbulence.**

**The paper covers a topic relevant for AMT. It is well structured. I recommend publication after the comments listed below are considered by the authors.**

We thank the reviewer for this general assessment of our manuscript and for his detailed comments which will help improve the text of our manuscript.

**Although well structured, many details of the paper were hard to follow. The clarity of the paper (e.g. wording, etc.) could be improved and the manuscript would benefit from a good editorial review. I did not include these types of suggestions in this review.**

We will carefully check and improve the grammar, spelling and clarity of our text.

**(1) Can the authors note on Figure 1 or somewhere else where the "true" releases were from (aka which star)? I notice that one release site is much further than the others. Is there anything about this location that follows through to the results (e.g. harder to pinpoint or quantify?).**

Only a few but not all of the actual locations of the releases were roughly indicated in Figure 1. The size of the ATEX zone (represented by the red line) in this figure was too small to get a precise view of these locations and we provided more detailed figures (like Figures 2 and 3) with a clearer view of the release locations for few releases only. Following this comment, we will clarify the legend of Figure 1 and provide Figure R1 in the supplementary material. This Figure R1 shows all the release locations in the ATEX zone (of note is that one release location can correspond to different releases since the same equipment was sometimes used for several releases).

The seven releases locations quite distant from the cluster of the other ones (corresponding to releases #5, 8, and 12-16) correspond to the larger emission rates $\geq 10$ g/s but not to higher errors in the release rate or location errors in a general way.

[Figure]

Figure R1: Release locations (red stars) in the ATEX zone. Different releases can have identical locations.

**(2) Can the authors further clarify why the Gaussian Plume model is a reasonable transport model for their study given some of the inherent drawback the authors note in paragraph 255. I found this discussion difficult to understand.**

We will extend this paragraph to clarify our points.

**Beyond the points that the authors list, isn't this application based on how field data is collected – given that the study has constant emissions for each release, are the measurement averaging time comparable to the source-to-receptor travel time?**

We are not sure about which "averaging time" the reviewer speaks about.

Getting some signal along the measurements timeseries means that the measurement period (which has started at the start of the release) exceeds the source-to-receptor travel time. We can add that the typical time between the first and last plume cross sections (when having more than two plume cross sections) is longer than the typical source-to-receptor travel time that we derive based on the average wind speed.

We did not try to average the concentration measurements 2D fields in time neither over the whole period of release or over a period adjusted to account for the "source-to-receptor travel time" because it clearly appeared that the whole release duration was too short and the number of plume cross-sections too small to get the convergence towards the average plume that one can expect under constant mean wind conditions but varying turbulent patterns. In practice, with only few (Np) plume cross-sections often away from each other, the average of the concentrations would have resulted in a Np peak or Np-modal pattern hardly comparable to the Gaussian model. This is why, instead of averaging the concentrations, we have compared the Gaussian model to each of these Np plume cross-sections.

**It seems like most of their reasons have to do with the fact that the alternatives are just too difficult to implement.**

This reading of this paragraph is a bit biased: only one of our four points really corresponds to this. The three others indicate that using more complex models would probably be useless (they would hardly allow for better results in the experimental conditions tackled during this campaign).

**How do expect the Gaussian plume model assumption to impact the uncertainties on their results?**

To our knowledge, it is very difficult to quantify the level of uncertainties associated with the comparison of the Gaussian model to instant plumes with turbulent patterns. We can claim that it is a source of random uncertainty but not of bias (since the Gaussian plume should simulate the average plume correctly). Tests with pseudo-data generated with CFD/LES models could feed such an assessment, but (1) it would provide typical (nearly qualitative) ideas about the level of uncertainties (2) it is out of the scope of this paper. We will now better stress that such a comparison between a Gaussian model and instant plume cross-sections is a source of uncertainty which is accounted for in the minimization of the cost function (and we will connect it to the discussion we already had about the "model error" in section 3.2).

**(3) Can the authors better explain how the authors used the Gaussian model to simulate mole-fractions (Am) and then use within the inversion to minimize the cost function?**

We will make some efforts to extend our explanations to make them clearer. In particular, we introduced a source of confusion by using the same notations ($x$, $y$, $z$) for the Gaussian model equation and for the potential source locations in the ATEX zone. We will now use different coordinate labels for these two different things.

We assume that this source of confusion is the main explanation for this general comment by the reviewer, since we think that all of the answers to the reviewer's questions within (3) were answered to in the original text. In particular, we believe that the content of the second review implies a clear understanding of these specific points.

Regarding this first question, section 3.1 explained how a Gaussian model is used to compute concentration fields (using its own $x$, $y$, $z$ coordinates), the second paragraph of section 3.2 explained how the direction of this model was set as that of the direction from a potential source location to the centre of the plume cross-sections, and the explanations below equations 4 indicated how Am is computed from the concentration field simulated with the model.

Regarding the use of the model to minimize the cost function, see our answers below.

**Where is xe, ye, and Qe specified in the equations? It seems like the authors varied the release rate (Qs) which I assume goes into the Gaussian model to create an ensemble of modelled concentrations (Am)?**

Yes, this was explained in the third paragraph of section 3.2: the inversion system iterates on a set of source location (defined by another coordinate system, different from that of the Gaussian plume model) and rates, and for each of them, it computes the concentration field with the Gaussian model, then Am and thus $J$. $x_e$, $y_e$ and $Q_e$ are the source location and rate which yield the smallest $J$, as was explained in the 4th paragraph of 3.2 (it was also an implicit information

from the 3rd paragraph). $z_e$ was fixed in the experiments shown in our manuscript as was explained in the 3rd paragraph of section 3.2.

**Then is ye is just the length of the centroid to the location of the observed mole fraction that with that minimizes the sum of the squared residuals for the two terms in equation 3?**

$y_e$ is not a length, $(x_e, y_e)$ is a position in the ATEX zone. It's definitely the one that, together with $Q_e$, minimizes $J$.

**Same with Qe?**

The system takes $(x_e, y_e, Q_e)$ as the set of parameters minimizing $J$ (the three parameters are varied independently to get this minimum).

**What type of minimization scheme did the authors use?**

Looping on all options for $(x, y, Q)$ as explained in the third paragraph of section 3.2.

**These sections need a lot more clarify.**

We are not really convinced by this statement but we will try to extend the explanations as mentioned above. We assume that the lack of clarity mainly came from the confusion between the two systems of coordinates $(x, y, z)$. As illustrated above, all the information and explanations were given in the original manuscript in a structured way. We have to be both clear and concise, and we should avoid to explain in details relatively simple things.

**For example, better nomenclature would help explain how the authors run their inversion with the plume model.**

Apart from using different letters for the different coordinate systems, we do not see which kind of nomenclature should be used. Projecting eq (1) into the system of coordinates used to locate the source in the ATEX zone would be useless and extremely complicated. Actually, many publications avoid to show equations such as equation (1) and just mention the use of a given type of model to simulate concentrations. We provided equations (1) in order to clarify the type of Gaussian model used and to document the specific way we set-up its dispersion parameters.

**A simple flow diagram would help here too.**

We think that the protocol is too simple to require a flow diagram. Explaining that

- we loop over all potential location and rate for the source to find the min for $J$

- we use a Gaussian model driven by the direction from the potential source locations to the plume cross-sections and by a potential rate to simulate the corresponding amplitude of these cross sections

should be enough.

**(4) Similar questions include: How many ensembles do the authors have for each Ao for a given grid locations?**

For each potential location, we simulate the full corresponding concentration field with the Gaussian model and the corresponding Am_i for all of the plume cross-sections during the release, as explained by the third paragraph of section 3.2.

**Their grid is 2,400 – I assume the authors used all of the Ao one-minute data within the "peaks" (obviously excluding those due to turns, weak winds, other criteria mentioned) or did the authors average in some way?**

We do not understand this question which is at odd with what the Ao correspond to (we do not understand what a "Ao one-minute data" can be). Ao_i correspond to the integral of concentrations above the background for the ith plume cross-section during a release, as was explained in the third paragraph of section 3.2. We will slightly extend the text to make this information easier to catch.

**How many Ao measurements did this amount to for each release? Are the number of Ao the same for both gases?**

Table 2 indicates the number of plume cross-sections (also called peak) Np and thus the number of Ao per release. It varies depending on the conditions for each release, and thus between different $CH_4$ releases as well as between $CH_4$ and $CO_2$ releases.

**All these points should be better clarified in the draft because it is very hard for the reader to follow the method, how it was applied, and thus be able to understand the results and discussions.**

We find this comment a bit excessive as indicated by our answers above. However, we will still extend paragraphs to dedicate more time to the different pieces of information/explanation and to better separate them.

**(5) It seems like one of the major assumptions of the work is that the authors provide the actual source height of each release to the inversion system. Later, the authors explain that this height might be artificially too low due to turbulence. I would recommend running a sensitivity test varying the source height. For one, it might show if the source height is effectively "higher", then it explains their results. It would also show the sensitivity of the results due to this assumption.**

Figure R2 shows an analysis of the sensitivity of the release rate estimate to the choice of the release height ($z_e$). The sensitivity analysis is illustrated for two releases (one of $CH_4$ and one of $CO_2$). The release height $z_e$ is varied from the actual source height to 12 meters with an incremental interval of 0.5 m. These figures show that the release rate estimations increase with the increase of the release heights. The lack of continuity in the curve for release-1 is due to the discretization in the set of release rates that are tested to find the optimal value (with a resolution of 0.05gCH4/s as detailed in section 3.2 and illustrated by Figure R2a)).

This behavior was expected and implicitly discussed in the 3rd paragraph of section 6 since the actual release height is close to or above the measurement height. In such a situation, increasing the release height increases the vertical distance between the measurements and the core of the Gaussian plume, and thus the amplitude of the modeled plume cross-section for a given release rate. Higher rates are needed to get a given amplitude with a higher release height. Therefore, we do not provide these results in the supplementary material of this paper.

However, we will mention the conclusion from such tests in section 6 since they demonstrate that at some stage, increasing the release heights can rapidly (after adding 4 meters for the release-8 as shown by figure R2b)) yield rate estimates larger than the actual release rates. In the absence of robust information on the injection height, the choice of higher release heights would be somewhat random and could yield larger biases than the negative one when using the height of the source.

It is important to note that these tests indicate that the estimate of the horizontal location of the release is weakly sensitive to the changes of release height.

The discussion on this topic will be extended to include some of these points in a more explicit way.

[Figure]

(a)                                        (b)

Figure R2: Estimates of the release rates with varying release heights for (a) release-1 ($CH_4$) and (b) release-8 ($CO_2$). The first estimates of the release rates correspond to the actual source heights. The dotted red lines represent the actual release rates.

**(6) It is unclear how/if the authors estimate uncertainties on their estimated parameters. Can the authors please provide more information? If the authors did not estimate them, I recommend a sensitivity analysis as noted above to help provide some measure of the variability of their estimates to assumptions. The authors must make some attempt at estimating uncertainties.**

The proposed inversion framework does not provide estimate of the uncertainties in the estimates of the release parameters. However, our inversions are applied to tests with controlled releases in which the true parameters are known: comparisons between estimates and true parameters is the best assessment of the uncertainties in the method. Based on this comparison, our discussion section discusses the uncertainties in the method. It is a common practice for sitescale inversion of pollutant emissions, an activity which allows for controlled emissions.

In many atmospheric inversion cases, the methods cannot be tested with controlled releases and different types of assessment of the uncertainties are needed to provide insights about the level of reliability. Statistical inversions or sensitivity tests can provide estimates of the uncertainties. However, they systematically rely on explicit or implicit assumptions about the sources of errors. In particular, critical sources of uncertainties can be missed and the characterization of the sources of errors cannot be perfect. We can list many sources of uncertainties here:

characterizing all of them and quantifying their impact properly with sensitivity tests would be quite complex and a full study in itself. But this is out of the scope of our study which is about testing and evaluating the technique based on controlled releases.

**(7) Can the authors provide a histogram (or something similar) to show how the 5th percentile value compares to the other Ao mole fractions for justification of background?**

The term "other Ao mole fractions" confuses the question but the following explanations clarify it.

**I would expect that the surrounding vegetation that the authors mention doesn't have too much impact given the short duration of their measurement time period but trucks passing by would. If these are just spikes in the data, I can imagine that the authors could just remove them.**

As an example, histograms of the measured concentrations at ~0.3-0.4 Hz frequency during one $CH_4$ (release-1) and one $CO_2$ (release-2) release are shown here in Figure R3 for the reviewer's reference. Similar types of histograms are also observed for other releases.

However, such histograms cannot help understand the variations of the background aside the plume cross-sections. These variations must be directly analyzed on series of measurements such as those shown in Fig 2(c).

As discussed in the manuscript, the baseline outside the plume cross-sections is smooth for the $CH_4$ releases, while it is noisy in some of the $CO_2$ releases. However, even for $CO_2$, we do not observe "spikes" in these timeseries. In any case, if infrequent, such spikes would hardly perturb the baseline during plume cross sections (or it could easily be detected in the shape of the plume cross-sections) nor enter into the computation of this baseline based on the 5th percentile of the values, which is the only thing that matters here. Therefore, there should not be any need to remove such spikes.

[Figure]

(a)                                             (b)

Figure R3: Histogram of the observed mole fractions at ~0.3-0.4 Hz frequency for (a) release-1 ($CH_4$) and (b) release-8 ($CO_2$).

**(8) The location error for CH4 is very large**

We discuss this point in section 6 and acknowledge it is large.

**especially given the size of the domain and potential release location sites (in general – how far apart are these?**

The distances between all these release points vary from ~2m to ~38m with an average distance of ~20m.

**Hard to tell from the schematic –**

As said earlier, we will now provide the figure R1 in supplementary material which is clearer about this.

**I would assume that the average spacing between these would act as a design requirement for the inversion).**

We do not agree since the inversion is not aware of the different locations from which the releases can be made. The "potential source locations" in the inversion cover the whole ATEX area.

**I suspect that this may have something to do with the fact that the inversion is very underdetermined. Is there any additional information that the authors could provide the system to help reduce the location errors?**

We invite the reviewer #1 to read the answer to the second review on this topic. In theory, the problem is not underdetermined. In practice, we did not find an easy way to constrain it fully properly to get precise location estimates. The discussions on this point and on further improvements to better constrain the problem will be extended in section 6.

**For example, the authors noted the potential release sites – if these are known a priori, the authors could limit their estimated locations to solely these grid cells.**

We can definitely decide what is the level of knowledge of the inversion on the release locations. However, our aim is to evaluate the potential of the method for realistic situations where the area of the fugitive leaks would be roughly known but where we would be looking for the precise location of these leaks. The knowledge which corresponds the best to such a situation during the controlled releases is limited to the information that the source is within the ATEX zone.

In practice, "blind releases" were conducted during the campaign to test systems for real situations of fugitive leaks: the participants knew that the releases occurred in the ATEX zone, but did not know about the list of potential point of releases shown in Figure R1 (nor about the release rates). Those are the conditions we have followed even after having obtained the locations and rates for all releases.

We will insert text regarding this discussion in section 3.2.

**At the very least, the authors could probably make some gross a priori assumptions about where they are not release locations to eliminate part of the solution space.**

If gross assumptions allow for excluding a part of the ATEX zone, then we have all reasons to assume that the inversion system will not estimate that the optimal location is in this part. Therefore, there is no need to exclude this part from the process before the inversion (there is no need to decrease the cost of the current computations).

**(9) Following (8) and (1) – can the authors say something about the location of the "potential" release points (some are bunched together while another is much further away and their results?**

See Figure R1 and our answers to (1) and (8).

**(10) I am not sure if averaging such a large variation of errors is really a representation of the expected errors – especially for only 7 samples. Can you justify?**

The result section discusses more deeply the distribution of errors. Furthermore, if speaking about the errors on the release rates, few of them exceed 30% so that the indication in the discussion and conclusion of the 20-30% average error can be viewed as conservative (the average is pulled upward by few large values) and makes sense.

**(11) Given the range of the release magnitudes – I think it would be nice to see some standard error statistics instead of relative error in the results.**

We do not understand this point. We would actually answer that because of the wide range of release rates, it makes more sense to analyze scores of relative rather than absolute errors.

The transport of $CH_4$ and $CO_2$ at the temporal and spatial scale of our experiments is linear, the precision of the instruments used to measure $CH_4$ and $CO_2$ is very fine, and the variations of the $CH_4$ and $CO_2$ are very low compared to the amplitude of the concentrations in the plume cross-sections. Therefore, in theory, the relative error should be the same whatever the actual release rate.

Making statistics with absolute errors would be roughly equivalent to focus on the results from the 45gCH$_4$/s release for $CH_4$ and from the two 150gCO$_2$/s releases for $CO_2$.

**(12) Why aren't there error bars on Figure 5?**

Because our final error bars arise from statistics over all these individual bars. See our answer to (6).

**Having CH4 stacked on CO2 begs for a comparison between the two but they aren't because releases go from 1-16. I would recommend putting these plots side by side instead.**

See our answer to (11): we disagree with this attempt at comparing absolute errors. For the sake of visibility, we thus prefer to keep the figure as it is.

**(13) Is one of the main reasons the authors used CO2 and CH4 is because of the background issue?**

Yes. The difference of precision of the $CH_4$ and $CO_2$ measurements should be another one even though these precisions are so fine that it plays a minor role in the resulting uncertainties.

**For replication sake, it would have been nice if the authors mimicked their release of CO2 and CH4 (aka same locations, magnitudes, durations, etc.) to be able to compare these. I understand that the authors cannot rerun the experiment but maybe the authors can group the releases into "similar" types between the two gases to help with the interpretation of results later on?**

The comparison between the relative error obtained for $CH_4$ and $CO_2$ should be the best way of comparing the impact of having different types of background and measurement precision for $CH_4$ and $CO_2$. See our answers to (11) and (12). However, the $CH_4$ and $CO_2$ releases occurred under very different wind conditions, which is the main driver of the differences between the results obtained for $CH_4$ and $CO_2$. Connecting large or small $CH_4$ and $CO_2$ releases would further artificially increase this effect of the limited sampling of wind conditions for each species. For these different reasons, we do not agree with such a suggestion.

**(14) What can the authors say about the practical application of their results? If the authors need 30 min of sampling – e.g. to improve their estimations – is this typical duration of an intermittent event?**

We think that these points were properly discussed in the introduction and in section 6 but we will slightly extend these sections to improve the clarity.

Saying 30 min releases would allow for more precise estimates does not imply that the results obtain with less than 10 min release are useless. 30% precision estimates of releases can be extremely useful in many cases. As said in introduction, all types of event can be encountered in industrial sites emitting $CH_4$.

**How many samples would the authors be able to get in a realistic event?**

This depends on the targeted sources. The duration of fugitive leaks can range from few seconds to more than 1 year. If turning the question into "How many samples would the authors need to get a good estimate" than answer would depend on the target precision. This targeted precision depends on the needs from end users.

Our reasoning goes the other way around: for the given emission duration, we provide a typical number of samples and a typical precision of the release rate estimates.

**The authors will always have a sampling bias during the day**

Why ? the instruments presented in this study can be operated during nighttime and we have Gaussian model formulations for nighttime conditions.

**so what does say about their methods?**

The topic of the need for continuous monitoring of the emissions and how the focus of our study fits into this more general topic was discussed in the introduction.

**How would the authors extend this for something more useful and practical for operators?**

We believe 30% precision instant estimates of releases from industrial sites are currently useful for operators, most of who do not know the typical magnitude of the emissions from their sites. As detailed in the introduction, the controlled releases were designed and organized by TOTAL in order to support the development of solutions for the sites they operate.

This study mainly aims at supporting the development of systems that could be used operationally. We believe that the test of methods and the results presented here go in this direction.

**(15) Again, I think presenting the results in relative errors is a bit misleading**

See our answers regarding this point.

**and I don't think that the authors demonstrated much skill in actually identifying the location of the leak – which is key for the application of this work.**

We think that we properly discussed this point in section 6 and acknowledge the limitation for the source location.

In addition to our previous answers regarding these points, we should mention that the target of the release rate when knowing the release location is a topic of many papers in this research domain and section 6 provided some positive comparisons to past studies regarding this.

**I don't think this warrants a rejection of the paper but a more realistic description of their results and methods.**

Given the various comments and questions from the reviewer and our answers to those, we do not understand the meaning of the term "realistic" in this sentence.

**I think that detecting and quantifying release may just require a better transport model – even if it is just slightly more sophisticated (e.g. AERMOD).**

This assumption is regularly raised but in practice, for experimental conditions similar to that presented here (in particular with flat terrain), one can hardly find publications demonstrating the need for using models more complex than Gaussian models. Furthermore, in practice, the systems commercialized for the monitoring of $CH_4$ emissions generally rely on mass balance approaches or Gaussian models (Fox et al, 2019; Mønster et al, 2019, etc.).

**(16) The authors suggestion of a hybrid approach is intriguing. Is there any other work that explored these ideas?**

We are not aware of such a hybrid approach in the literature. This idea follows the rationale we develop in section 6. We wish to explore this in near future.

**References**

Fox, T. A., Barchyn, T. E., Risk, D., Ravikumar, A. P., & Hugenholtz, C. H. (2019). A review of close-range and screening technologies for mitigating fugitive methane emissions in upstream oil and gas. Environmental Research Letters, 14(5), 053002.

Mønster, J., Kjeldsen, P., & Scheutz, C. (2019). Methodologies for measuring fugitive methane emissions from landfills–A review. Waste Management, 87, 835-859.

---

## Author Comment (AC2) · 20 Jan 2021

We thank Joseph Pitt for his useful discussion on the principle of our inversion approach and on the use of plume widths as a further constraint on the release locations. These will lead to clarifications and to the extension of our discussions.

**This study presents a new approach to determine the location and rate of point source emissions, and tests the method using a series of controlled releases. Mobile measurements of mole fraction are made on board a car, which performs repeated transects through the plume downwind of the release. A Gaussian plume model (driven by local meteorological measurements) is used to simulate mole fractions at the measurement locations for an ensemble of release locations and emission rates. The offset between modelled and measured locations of the plume centre is calculated, in addition to the difference between the integrated plume enhancement (a.k.a. plume amplitude) for the modelled and measured datasets. Estimates for both release location and emission rate are derived by minimising a cost function that seeks to reduce these two measures of model-measurement mismatch.**

**The study is well-motivated and the details of the experiment are clearly described.**

We thank J. Pitt for this general and positive assessment of our study.

**Unfortunately, if I've understood it correctly, I think there is a fundamental problem with the method developed here. As far as I can tell, there is insufficient information content in the plume amplitude and plume-centre location to constrain both the location and rate of emissions.**

We feel that this general comment confuses two different things

- 1) the idea that there could be a lack of information to constrain both the location and rate of the emissions in the theoretical frame of the proposed method

- 2) the lack of information to constrain both the location and rate of the emissions in practice, when applying the method to the specific experiments with the specific configurations presented in this study

We do not agree with (1) which is the basis of this general comment, but our analysis and discussions support (2).

Actually, this general comment

- nearly follows some of the point of the discussion we led in the second paragraph of section 5.3 (lines 469-475)

- misses the follow-up of this discussion in the third and sixth paragraph of section 6

- extrapolates what is actually a diagnostic from the first set of experiments into the assumption that it could be a fundamental problem of the method which could have been anticipated (an assumption we disagree with)

However, we realize that by having applied our method to a release for which we had one plume transect only without any warning, we have raised a source of confusion regarding its concept. In our revised manuscript, we will extend our discussions on the topic (in particular at the beginning of section 5.3 and in section 6) to clarify it. We will also explain why the case with one plume transect only is analyzed and used for the general statistics of uncertainties in the

release estimates, even though our method relies on the use of multiple plume transects to derive the location of the sources.

See our detailed answers below.

**It seems like the plume-centre location can be used to constrain the location of the release to a line along the average wind vector,**

Not when considering two or more plume transects which is the fundamental basis of the method.

**while the plume amplitude can either constrain the emission rate for a given release location on this line, or the release location for a given emission rate. Take the following example where the wind is perpendicular to the transects:**

[Figure]

**In this case any release location along the dotted line will result in the same modelled plume-centre location. But a source at release point 2 with a low emission rate will produce the same plume amplitude as a source at release point 1 with a high emission rate. These would produce the same value of J for this transect.**

These considerations regarding the lack of constraint on the release location when having one plume transect only are misleading. A major cause for such considerations was our analysis of a release with one plume transect only without raising any warning. However, the principle of our method relies on multiple plume transects.

The additional explanation will be added to the description of the method in section 3.2. And an explanation regarding the analysis of release #12 will now be given in section 4.2.

**In all but one case presented in this study there are multiple transects of the plume.**

Yes, it is the critical point.

**This adds extra information to the case above. But I don't think it is being used to constrain the location in a useful way. Because the emission rate Qe does not impact JW, J is minimised by setting the location of the release to minimise JW, then setting the release rate to minimise Jp given this location.**

The minimization is not iterative, it minimizes both components simultaneously.

**If the plume centre-location is different for two transects then JW is minimised by moving the source location further away from the transects.**

In principle, such a tendency should be balanced by the need to fit the individual areas of the plume transects. However, our results (and our discussion) showed that, in practice, Jw is much larger than Jp, which explains that, yes, the source location is pushed away.

**Figure 3 is the perfect example of this in action. JW alone sets the source location, but the x-value of this source location is purely an artefact of the way the cost function has been constructed.**

See our explanation above.

**In section 5.3 it is stated that Jp does not "push far enough for finding a source location".**

The idea of using $J^{log}$ (section 5.3) was driven by the need to overcome this problem, adding constraint on the fit to the amplitude of each plume transect. We will now extend the discussion at the beginning of section 5.3.

**But in most cases it has no impact on the estimated source location at all, I suspect for the reasons outlined above.**

We do not see any explanation above in the reviewer's comment regarding why Jp would not push far enough to find the source location. Our explanation is the relative weight between Jp and Jw, not the principle of the estimation method. Several investigations discussed in section 6 could help overcome this issue. We will extend these discussions in the revised manuscript.

**This is apparent from tables 3 and 5 – the locations are usually the same regardless of whether J or Jlog is used as the cost function, because in both cases JW is the same.**

We thought that using $J^{log}$ would solve for the problem by putting more weight on the fit to "small" plume cross-sections. However, $J^{log}_p$ kept on being much smaller than Jw. This led to our discussions on that topic in section 6.

**In cases where there is a difference I guess that it probably arises from some combination of the geometry of the ATEX zone boundary and the discretisation of release locations and emission rates.**

The discretization of the release locations and rates is kept the same when using J or $J^{log}$, so we do not really understand this assumption. On the opposite, our assumption that using $J^{log}$ puts more weight on the fit to small plume transect properly explains why the location errors are different when using it instead of J.

Note that this result demonstrates that Jp or $J^{\log}_p$ does have some weight on the estimation of the release location.

**It's entirely possible that I've misunderstood what's going on here – if so then I'm sure the authors can put me straight! But until I have faith in the overall approach, I can't recommend that this paper is published. If the authors can convince me that the method is sound then I'm happy to provide more detailed feedback on specific points. Otherwise I think the best option might be to reject this paper in its current form and consider what additional information could be used to better constrain the problem.**

We hope that our answers clarify this whole discussion.

**One obvious candidate would be to use the plume width in some capacity, but I think that to do so would require a more complex model,**

Yes, it would.

**as one would need to simulate the likely width of the instantaneous plume (rather than the time-averaged plume represented in the Gaussian plume model). Perhaps that is a bad idea…**

It is a good idea and definitely something to tests with a more complex model. We will add a small discussion on that point in section 6.

**but either way I think some additional constraint is required in order to render this approach useful in determining source location as well as emission rate.**

Other options to solve for this issue when applying the method to our specific study case will be better stressed in section 6.

---

## Referee Report (RR1)

Thanks to the authors for further explaining their approach. I can now see that with multiple transects this method can in theory constrain the location as well as the strength of the emission. I had missed the important point that the X and Y axis in Eq. 1 are defined according to $\theta_m$ rather than $\theta$. Consequently, while $J_w$ will always be minimised by moving the source further away, in cases where the plume amplitude differs significantly between transects $J_p$ could have lower values for release locations closer to the transect. So I agree with the conclusions of the authors: in theory there could be enough information to constrain the location and emission rate using this approach, but in practice it has not worked in this case. I think that conclusion is a useful one, so I suggest that this paper should be published in AMT, but I have some additional comments that I think it would be good to address.

I now see how the relative amplitude of plumes on different transects could in theory provide a constraint on the location of the source. However, if I have understood this correctly, this constraint only exists when the wind direction has a significant component parallel to the transect. To demonstrate what I mean, consider the two diagrams below. In the first the wind is perpendicular to the transect, while in the second there is non-negligible parallel wind component. Both diagrams represent a top-down view of the area, with the path of the measurement vehicle in red and the measured $CH_4$ enhancements along the transect shown right.

[Figure]

[Figure]

In diagram 2), where there is a significant component of the wind parallel to the transect, the plume is measured closer to the source on transect 2 than on transect 1. Consequently the plume amplitude is larger on transect 2, and the relative amplitude of the plume on transect 2 vs transect 1 can (in theory) help to constrain the emission source location.

However, in diagram 1), where the wind is perpendicular to the transect, the plume amplitudes are the same on transect 1 and transect 2. In this case there is insufficient information to constrain the location of the source, even in theory, because the plume amplitudes could be equally well simulated by various combinations of emission rate and source location. Because $J_w$ will always be minimised by increasing the distance between the source and the measurements, the estimated emission location will be pushed further away from the transects.

So it would seem that to give this method the best chance of working, one would want to conduct sampling in the wind conditions shown in diagram 2) and avoid the perpendicular wind direction shown in diagram 1). If this is the case then I think this is a conclusion worth highlighting for future studies. From table 2 of the manuscript it seems that sampling took place under a variety of wind directions, allowing this hypothesis to be investigated. I have two suggestions for this investigation:

1. Include plots of the form of Figure 3 for all releases in the SI. This will demonstrate whether the relative plume amplitude on different transects does sometimes constrain the source location (using the current definition of J), or whether the estimated source location is pushed to the edge of the box in all cases

2. Test the impact of varying the plume amplitude uncertainty in $J_p$. Currently a 100% uncertainty in modelled plume amplitude is assumed – as mentioned in the discussion, lower values for this uncertainty could help to constrain the source location. It would be useful to test various choices of this parameter (e.g. 80%, 60%, 40%, 20%, 10%). In cases where there is a component of the wind parallel to the transect, there may be a value below which this constraint kicks in and the location estimate is no longer forced to the furthest distance from the transects. Clearly this would not mean that a lower uncertainty is justified, but it would give us a sense of the model accuracy that would be required for the successful application of this method. This threshold accuracy would presumably be a function of wind direction (relative to the transect). If conducting this analysis for all releases is not feasible, it would at least be good to see these results for a selection of releases covering different wind conditions.

I appreciate that my suggestion 2 has some overlap with the current analysis, where $J^{log}$ is used in place of J. But I think it would be useful for future studies to include an estimate of the required model accuracy, even if it is only strictly applicable to the conditions encountered during these controlled release experiments.

Finally, if in this study the cost function was always minimised by placing the source location at the furthest point from the transect, then I think the comparison of estimated emission rates to other studies should be done using the fixed-location results. As the authors point out, the tendency to overestimate the distance to the source partially counterbalances the tendency of the inversion to underestimate emissions. Presumably if you increased the size of the grid then the location error would be larger and the emission rate error smaller, but this would not mean that the accuracy of the method at estimating the emission rate was inherently improved. I think it is useful to separate the two issues (location and emission rate) in the discussion section; first discuss what improvements would enable the method to estimate the location of the emission with some skill, then discuss the accuracy of the fixed-location emission rate estimates relative to other studies.

In addition to my general suggestions above, I have a couple of specific comments:

- Why was $\theta_m$ set to $\theta \pm 2\sigma_\theta$ when it was outside this range? Surely if the angle between a potential release location and the observed plume is very different to the measured wind direction then that location is unlikely to be the source of the release? Therefore it seems reasonable that $J_w$ will be very large for such a location, and it is not clear to me why it needs to be limited in this way.
- It would be useful to include the results of the fixed-location experiments (using both J and $J^{log}$), either in tables 3 and 4 or in the SI.

---

## Referee Report (RR2)

I want to thank the authors for their patience in explaining to me the principal on which this method is based. I can see that the example I gave in my previous review (with two mirror-image transects) represents a rather unique case for which this method would not work, and that in theory it could work under more general conditions where the mean wind during the release was perpendicular to the transects. The changes made to the text and the additional SI figures have also helped to clarify this. I suggest that this revised manuscript should be published in AMT.

As future work, it could be interesting to use a CFD model to simulate some other possible releases, then test how many transects would be required for this Gaussian plume inversion method to accurately estimate the release rate and location. Anyway that's just an idea – it's clearly well beyond the scope of this study and would involve a lot of work!

---

## Author Response (AR2)

**We thank Joseph Pitt for his series of reasoning on our inversion approach and results. His new suggestions have helped, again, to refine our analysis and discussions.**

Thanks to the authors for further explaining their approach. I can now see that with multiple transects this method can in theory constrain the location as well as the strength of the emission. I had missed the important point that the X and Y axis in Eq. 1 are defined according to $\theta_m$ rather than $\theta$. Consequently, while $J_w$ will always be minimised by moving the source further away, in cases where the plume amplitude differs significantly between transects $J_p$ could have lower values for release locations closer to the transect. So I agree with the conclusions of the authors: in theory there could be enough information to constrain the location and emission rate using this approach, but in practice it has not worked in this case. I think that conclusion is a useful one, so I suggest that this paper should be published in AMT, but I have some additional comments that I think it would be good to address. I now see how the relative amplitude of plumes on different transects could in theory provide a constraint on the location of the source. However, if I have understood this correctly, this constraint only exists when the wind direction has a significant component parallel to the transect. To demonstrate what I mean, consider the two diagrams below. In the first the wind is perpendicular to the transect, while in the second there is non-negligible parallel wind component. Both diagrams represent a topdown view of the area, with the path of the measurement vehicle in red and the measured $CH_4$ enhancements along the transect shown right.

[Figure]

[Figure]

In diagram 2), where there is a significant component of the wind parallel to the transect, the plume is measured closer to the source on transect 2 than on transect 1. Consequently the plume amplitude is larger on transect 2, and the relative amplitude of the plume on transect 2 vs transect 1 can (in theory) help to constrain the emission source location.

However, in diagram 1), where the wind is perpendicular to the transect, the plume amplitudes are the same on transect 1 and transect 2. In this case there is insufficient information to constrain the location of the source, even in theory, because the plume amplitudes could be equally well simulated by various combinations of emission rate and source location. Because $J_w$ will always be minimized by increasing the distance between the source and the measurements, the estimated emission location will be pushed further away from the transects.

So it would seem that to give this method the best chance of working, one would want to conduct sampling in the wind conditions shown in diagram 2) and avoid the perpendicular wind direction shown in diagram 1). If this is the case then I think this is a conclusion worth highlighting for future studies.

**Two processes drive the amplitude of plume transects perpendicular to the wind at fixed height above the ground as a function of the distance from the source (for given wind speed and direction): the plume amplitude is smaller at larger distance due to (i) the loss of larger tails of the plume when integrating between the "edges of the observed peak" because of its wider extent and smoother shape (ii) the larger vertical mixing decreasing**

the concentrations close to the ground. The (iii) change of angle between the wind direction and the plume transect adds to the variations of the amplitude of the plume transects.

Of note is that the modeling framework is driven by the effective wind direction corresponding to the direction from the source to the observed plume transect (see J. Pitt's own statement above: "the X and Y axis in Eq. 1 are defined according to $\theta_m$ rather than θ "), so that a plume transect perpendicular to the measured wind is hardly perpendicular to such effective wind directions.

We are not sure about how to interpret diagrams 1) and 2). In 1) transects 1 and 2 are represented at the same distance from the source with the same angles between the corresponding effective wind direction and the line of measurement. This would lead to the same plume amplitude, but the reason would not be that the measured wind is perpendicular to the line of measurements. If shifting transects 1 and 2 in a dissymmetric way along the line of measurements, the three processes (i-iii) described above would lead to different plume amplitudes as for the two transects in 2).

Therefore, we do not agree with this general reasoning. We add that plume transects perpendicular to the effective wind are actually preferable since providing clearer limits for the plume, and since being less prone to large errors due to uncertainties in the wind direction.

We insert in the manuscript part of the clarifications given in this answer.

From table 2 of the manuscript it seems that sampling took place under a variety of wind directions, allowing this hypothesis to be investigated. I have two suggestions for this investigation:

We do not agree with the previous reasoning but we still consider the following suggestions since they can help characterize the behavior of the inversion.

1. Include plots of the form of Figure 3 for all releases in the SI. This will demonstrate whether the relative plume amplitude on different transects does sometimes constrain the source location (using the current definition of J), or whether the estimated source location is pushed to the edge of the box in all cases

The figures are now included in the SI (Figures S2-S17). They reveal that $J_p$ is very smooth, and, if ignoring the bounds of the ATEX area, it would have its minimum outside this area. Consequently, minimizing $J_p$ leads to locate the source on a border of the ATEX zone too (most of the time on a border different from that where the minimization of $J_w$ pushes the source). One explanation is the lack of plume transects for constraining the computation. For example, if having two plume transects only, an infinity of solution can lead to the respective amplitude of these transects. Even when having more plume transects, $J_p$ may tend to be driven by a subset of transects, which is another reason for having tried to work with $J^{log}$. The idea of having a cost function $J$ combining $J_w$ and $J_p$ was partly to overcome such a limitation.

Again, we expand the explanations in the manuscript to better clarify these considerations.

2. Test the impact of varying the plume amplitude uncertainty in $J_p$. Currently a 100% uncertainty in modelled plume amplitude is assumed – as mentioned in the discussion, lower values for this uncertainty could help to constrain the source location. It would be useful to test various choices of this parameter (e.g. 80%, 60%, 40%, 20%, 10%). In cases where there is a component of the wind parallel to the transect, there may be a value below which this constraint kicks in and the location estimate is no longer forced to the furthest distance from the transects.

**We have conducted these tests and the summary of the results is provided in SI (Figures S18 & S19). $J_w$ has been reweighted by a factor λ in $J$ and $J^{log}$ which is equivalent (via the the division of the resulting $J$ by λ) to consider a relative model error of $\sqrt{\lambda}$ when modeling the plume area A in $J_p$ or when modeling log(1+A) in $J_p^{log}$. Surprisingly, on average, the smallest location error is generally found for λ=1 or 0, so that the location error is hardly smaller than that with our default inversion configuration. But for $CO_2$ release inversions when minimizing $J$ and $J^{log}$, the optimal average location error can be found for λ=1.6% (i.e. a relative model error of ~13%) and λ=0.4% (i.e. a relative model error of ~6%), respectively. Furthermore, the curves of average location errors as a function of λ for $CH_4$ releases when minimizing $J$ or $J^{log}$ have local minima with values close to the optimal one obtained for λ=1 (for λ=0.016, i.e. a relative model error of ~13%, for $J$ and 0.08, i.e. a relative model error of ~9%, for $J^{log}$). With such values, some of the releases are located well inside the ATEX zone (see SI Figures S20 and S21 for release no. 2). However, most of the release locations keep on being pushed against the border of this area since the resulting $J$ and $J^{log}$ functions keep on being quite smooth. We assume, again, that the lack of plume transects coupled to the model error explain it and the fact that the system misses the actual release location which should correspond to a local minimum of the cost functions.**

**We now discuss these results in section 5.3.**

Clearly this would not mean that a lower uncertainty is justified, but it would give us a sense of the model accuracy that would be required for the successful application of this method.

**We are ready to agree with the idea that it might give us "a sense" of the required model accuracy but the situation seems too complex for us to get robust insights on it and to discuss it correctly in one or two sentences. $J_p$ should balance the misfits and the model error, and normally, the two terms should be strongly correlated. Furthermore, here the requirements strongly depend on the formulation of $J_w$ and, strictly speaking, the optimal values of λ for the error location is 1 for the $CH_4$ releases (which corresponds to our standard set-up). We thus prefer to avoid discussing this specific idea in the manuscript but we add few sentences on the topic. Of note is that the results from the analysis when reweighting $J_w$ lead us better identify the potential need to derive model errors that are specific to the different plume transects.**

This threshold accuracy would presumably be a function of wind direction (relative to the transect). If conducting this analysis for all releases is not feasible, it would at least be good to see these results for a selection of releases covering different wind conditions.

**We have the results for all releases, but the problem appears to be too complex to derive such a general understanding of the behavior as a function of the wind conditions. Also, see our answer to the general comment above.**

I appreciate that my suggestion 2 has some overlap with the current analysis, where $J^{log}$ is used in place of J. But I think it would be useful for future studies to include an estimate of the required model accuracy, even if it is only strictly applicable to the conditions encountered during these controlled release experiments.

**We hope that our answers above clarify the reason why we prefer to avoid discussing the "required model accuracy".**

Finally, if in this study the cost function was always minimised by placing the source location at the furthest point from the transect, then I think the comparison of estimated emission rates to other studies should be done using the fixed-location results. As the authors point out, the tendency to overestimate the distance to the source partially counterbalances the tendency of the inversion to underestimate emissions. Presumably if you increased the size of the grid then the location error would be larger and the emission rate error smaller, but this would not mean that the accuracy of the method at estimating the emission rate was inherently improved. I think it is useful to separate the two issues (location and emission rate) in the discussion section; first discuss what improvements would enable the method to estimate the location of the emission with some skill, then discuss the accuracy of the fixed- location emission rate estimates relative to other studies.

**We agree with this point: we now better highlight the results from the experiments in which the release is fixed to its actual location in sections 5 and 6 and by revising the ranges summarizing the precision for the release rate estimates.**

In addition to my general suggestions above, I have a couple of specific comments:

- Why was $\theta_m$ set to $\theta \pm 2\sigma_\theta$ when it was outside this range? Surely if the angle between a potential release location and the observed plume is very different to the measured wind direction then that location is unlikely to be the source of the release? Therefore it seems reasonable that $J_W$ will be very large for such a location, and it is not clear to me why it needs to be limited in this way.

**Following this comment, we re-assessed the relevance of setting $\theta_m$ to $\theta \pm 2\sigma_\theta$ maximum, and we decided to remove this strong constraint. All the results have been updated. In a general way, they look very similar after this revision and this does not impact our analysis.**

- It would be useful to include the results of the fixed-location experiments (using both J and $J^{log}$), either in tables 3 and 4 or in the SI.

**We agree: we have included them in both tables 3 and 4.**